# Learning Adaptive Distribution Alignment with Neural Characteristic Function for Graph Domain Adaptation

**Wei Chen[1], Xingyu Guo[1], Shuang Li[1,†], Zhao Zhang[2], Yan Zhong[3],**

**Fuzhen Zhuang[1,4,†], Deqing Wang[2]**

[1]School of Artificial Intelligence, Beihang University, Beijing, China
[2]School of Computer Science and Engineering, Beihang University, Beijing, China
[3]School of Mathematical Sciences, Peking University, Beijing, China
[4]Zhongguancun Laboratory, Beijng, China
{chenwei23,shuangliai,zhuangfuzhen}@buaa.edu.cn

## Abstract

Graph Domain Adaptation (GDA) transfers knowledge from labeled source graphs to unlabeled target graphs but is challenged by complex, multi-faceted distributional shifts. Existing methods attempt to reduce distributional shifts by aligning manually selected graph elements (e.g., node attributes or structural statistics), which typically require manually designed graph filters to extract relevant features before alignment. However, such approaches are inflexible: they rely on scenario-specific heuristics, and struggle when dominant discrepancies vary across transfer scenarios. To address these limitations, we propose **ADAlign**, an Adaptive Distribution Alignment framework for GDA. Unlike heuristic methods, ADAlign requires no manual specification of alignment criteria. It automatically identifies the most relevant discrepancies in each transfer and aligns them jointly, capturing the interplay between attributes, structures, and their dependencies. This makes ADAlign flexible, scenario-aware, and robust to diverse and dynamically evolving shifts. To enable this adaptivity, we introduce the Neural Spectral Discrepancy (NSD), a theoretically principled parametric distance that provides a unified view of cross-graph shifts. NSD leverages neural characteristic function in the spectral domain to encode feature-structure dependencies of all orders, while a learnable frequency sampler adaptively emphasizes the most informative spectral components for each task via minimax paradigm. Extensive experiments on 10 datasets and 16 transfer tasks show that ADAlign not only outperforms state-of-the-art baselines but also achieves efficiency gains with lower memory usage and faster training. Code is available at: https://github.com/gxingyu/ADAlign.

## 1 Introduction

Graph-structured data (Shao et al., 2024; Zhang et al., 2024; Chen et al., 2024; Yuan et al., 2025; Chen et al., 2025b), characterized by intricate dependencies among nodes and edges, often undergoes substantial distribution shifts across domains (Li et al., 2022a; Guo et al., 2024). Such shifts break the inductive assumptions underlying conventional graph neural networks (GNNs) (Liu et al., 2021), leading to severe performance degradation when key attributes or topological relations vary. To mitigate this problem, graph domain adaptation (GDA) (Wu et al., 2023; Liu et al., 2024b) has emerged as a promising paradigm, aiming to learn domain-invariant representations that transfer knowledge from source to target domains and improve generalization across heterogeneous graphs.

In GDA, a popular line of work is cross-graph feature alignment (Chen et al., 2019; You et al., 2023; Liu et al., 2024b;a; Shao et al., 2024; Chen et al., 2025a), which directly minimizes distributional distances, such as KL divergence, Maximum Mean Discrepancy (MMD), or Wasserstein distance

---

†Corresponding authors: Shuang Li and Fuzhen Zhuang.

(Panaretos & Zemel, 2019), to reduce gaps between source and target graphs, offering both interpretability and empirical effectiveness. Early approaches typically employed a GNN to encode node attributes and structural features into unified embeddings, which were then aligned across domains using these distance metrics. However, alignment at this coarse global level often overlooks fine-grained distributional shifts, resulting in limited adaptation and increased risk of negative transfer.

Recently, several studies (Fang et al., 2025b;a; Yang et al., 2025) have decomposed graph distribution shifts into distinct components (e.g., node attributes, degree distributions, or graph homophily) and designed specialized adaptation strategies for each, offering a more fine-grained understanding of the factors hindering cross-domain generalization.

Nevertheless, these approaches often rely on heuristic strategies that first **manually design** graph filters to extract relevant features (such as node attributes or structural statistics) before performing alignment for separable distributional shifts. This process overlooks a fundamental property of real-world graphs: the dominant sources of discrepancy vary across transfer scenarios, with their interplay being inherently complex and unpredictable. For example, as shown in Figure 1, we visualize the distributional discrepancies across the three Airport (USA, Brazil, Europe) (Ribeiro et al., 2017) transfer scenarios, focusing on the top 5 features. We observe that the dominant sources of discrepancy vary across these scenarios. For instance, in the B-E scenario, features 2 and 3 exhibit the largest discrepancies, whereas in the U-E scenario, the features with the most notable shifts simultaneously change to 1, 2, and 4. Existing methods, however, often rely on fixed strategies that align only a limited set of these features, which makes them inadequate for capturing the full range of distributional shifts and the interplay between multiple factors. This underscores the need for adaptive approaches that can jointly capture and prioritize shifting dimensions, facilitating better alignment across transfer scenarios.

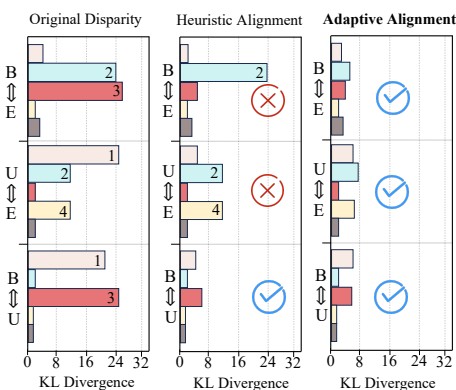

Figure 1: Distributional disparities across different scenarios. Each dimension (e.g., 1, 2) corresponds to a PCA-reduced feature, and the value represents the KL divergence between pairs of the top 5 features, computed for each transfer task, e.g., KL(B1||U1).

To address these challenges, we introduce **ADAlign**, an Adaptive Distribution Alignment framework for GDA. Unlike heuristic methods that rely on predefined alignment criteria, ADAlign automatically detects the most relevant sources of discrepancy in each transfer scenario and aligns them jointly, capturing the complex interplay between graph attributes, structures, and their dependencies. This makes ADAlign flexible and capable of handling a wide range of dynamic shifts across domains. To enable this adaptivity, we propose the Neural Spectral Discrepancy (NSD), a novel parametric distance specifically designed for graphs. NSD uses neural characteristic function (Ushakov, 1999; Ghahramani, 2024) in the spectral domain to capture feature-structure dependencies at multiple levels. A learnable frequency sampler adaptively prioritizes the most informative spectral components for each task within a minimax optimization framework. This approach offers a unified and tractable framework to quantify composite shifts, enabling alignment that is both theoretically grounded and adaptively sensitive to the nature of the transfer scenario.

Our contributions are threefold:

**(i)** We propose an adaptive framework that automatically identifies and aligns the most relevant sources of discrepancy in each transfer scenario, enabling flexible and task-aware adaptation to dynamic graph shifts.

**(ii)** We introduce NSD, a novel parametric distance for graphs that leverages neural characteristic function in the spectral domain to capture multi-level feature-structure dependencies, providing a unified approach to quantifying distributional shifts.

**(iii)** Extensive experiments on 10 datasets and 16 transfer tasks show that ADAlign outperforms state-of-the-art baselines, significantly reducing memory consumption and training time.

## 2 RELATED WORK

**Graph Domain Adaptation.** GDA (Zhuang et al., 2020; Li et al., 2025) aims to transfer knowledge from a labeled source graph to an unlabeled or sparsely labeled target graph, mitigating the challenges introduced by distribution shifts in graph-structured data. Early approaches (Ma et al., 2019; Wu et al., 2020; Zhang et al., 2021; Dai et al., 2022; Pang et al., 2023; Liu et al., 2024a) typically employed graph neural networks (GNNs) to encode node attributes and structural information into low-dimensional embeddings, and then aligned source and target representations using statistical discrepancy measures such as KL divergence (Menéndez et al., 1997; Johnson et al., 2001), MMD (Arbel et al., 2019), or Wasserstein distance (Panaretos & Zemel, 2019). Although progressive, these methods struggle to capture fine-grained, property-specific differences, often leading to suboptimal alignment and degraded performance when target graphs exhibit complex distributional shifts. Subsequent studies (You et al., 2023; Liu et al., 2024b; Fang et al., 2025b;a) have dissected graph distribution shifts into factors such as attribute misalignment, degree divergence, and variations in homophily, inspiring targeted adaptation strategies tailored to specific types of shifts. Despite these advances, most existing methods still rely on heuristic metrics, which remain insufficient for modeling the entangled, multi-level nature of graph discrepancies. In this work, we propose an adaptive distribution-alignment framework that holistically accounts for composite shifts.

**Characteristic Function.** Characteristic Function (CF) (Pólya et al., 1949; Yu, 2004; Ghahramani, 2024) establish a one-to-one correspondence between probability distributions and their Fourier-domain representations (Ozaktas & Aytür, 1995), providing compact yet complete characterizations that have motivated applications across machine learning. They have been widely used in areas such as generative modeling (Li et al., 2017; 2020; 2022b; Lou et al., 2023), where they measure gaps between real and synthetic data, and knowledge distillation (Wang et al., 2025), where their compactness enables efficient transfer of knowledge from teacher to student models. Despite these advances, the use of CF in domain adaptation remains limited, especially for graph-structured data. Given the complexity of graph distributions, where attribute, structural, and topological shifts are often entangled, CF offer a promising foundation for developing flexible adaptation frameworks.

## 3 PRELIMINARIES

**Notation**. We consider an undirected graph $\mathcal{G} = \{\mathcal{V}, \mathcal{E}, \mathcal{A}, \mathcal{X}, \mathcal{Y}\}$, where $\mathcal{V}$ and $\mathcal{E}$ denote the sets of nodes and edges, respectively. The graph structure is encoded by the adjacency matrix $\mathcal{A} \in \mathbb{R}^{N \times N}$, where $N = |\mathcal{V}|$ is the number of nodes. Each entry $\mathcal{A}_{ij} = 1$ indicates the presence of an edge between nodes $i$ and $j$, while $\mathcal{A}_{ij} = 0$ indicates no edge. Each node $v \in \mathcal{V}$ is associated with a feature vector, organized in the feature matrix $\mathcal{X} \in \mathbb{R}^{N \times d}$, where $d$ is the feature dimension. The node labels are represented by $\mathcal{Y} \in \mathbb{R}^{N \times C}$, with each node belonging to one of $C$ classes. Following prior works (Liu et al., 2024b; Huang et al., 2024; Fang et al., 2025a), we consider the unsupervised setting with two graphs: a source graph $\mathcal{G}^{\mathsf{S}} = \{\mathcal{V}^{\mathsf{S}}, \mathcal{E}^{\mathsf{S}}, \mathcal{A}^{\mathsf{S}}, \mathcal{X}^{\mathsf{S}}, \mathcal{Y}^{\mathsf{S}}\}$ and a target graph $\mathcal{G}^{\mathsf{T}} = \{\mathcal{V}^{\mathsf{T}}, \mathcal{E}^{\mathsf{T}}, \mathcal{A}^{\mathsf{T}}, \mathcal{X}^{\mathsf{T}}\}$, and focus on the node classification task (Xiao et al., 2022), where the goal is to train a GNN classifier $h$ that accurately predicts node labels $\mathcal{Y}^{\mathsf{T}}$ for the nodes in graph $\mathcal{G}^{\mathsf{T}}$.

**Characteristic Function.** The CF (Waller et al., 1995) is a fundamental tool in probability theory that represent a random variable in the frequency domain and provide a convenient way to analyze and compare distributions (Xie et al., 2022). For a random vector $\mathbf{X}$, the CF is defined as

$$\Psi_{\mathbf{X}}(\mathbf{t}) = \mathbb{E}\left[e^{i\mathbf{t}^{\top}\mathbf{X}}\right] = \int_{\boldsymbol{x}} e^{i\mathbf{t}^{\top}\boldsymbol{x}} dF_{\mathbf{X}}(\boldsymbol{x}), \tag{1}$$

where $\mathbf{t} \in \mathbb{R}^m$ is the frequency vector, $\boldsymbol{x} \in \mathbb{R}^m$ is a realization of $\mathbf{X}$, $F_{\mathbf{X}}(\boldsymbol{x})$ is the cumulative distribution function (Drew et al., 2000), $\top$ denotes transpose operator and $i = \sqrt{-1}$ is the imaginary unit. To capture oscillatory strength and directional shifts, the CF can be expressed in polar form:

$$\Psi_{\mathbf{X}}(\mathbf{t}) = |\Psi_{\mathbf{X}}(\mathbf{t})|\, e^{i\theta_{\mathbf{X}}(\mathbf{t})}, \tag{2}$$

where $|\Psi_{\mathbf{X}}(\mathbf{t})|$ denotes amplitude and $\theta_{\mathbf{X}}(\mathbf{t})$ the phase. This amplitude-phase representation provides intuitive insights into structural and geometric properties of distributions (Chen et al., 2021).

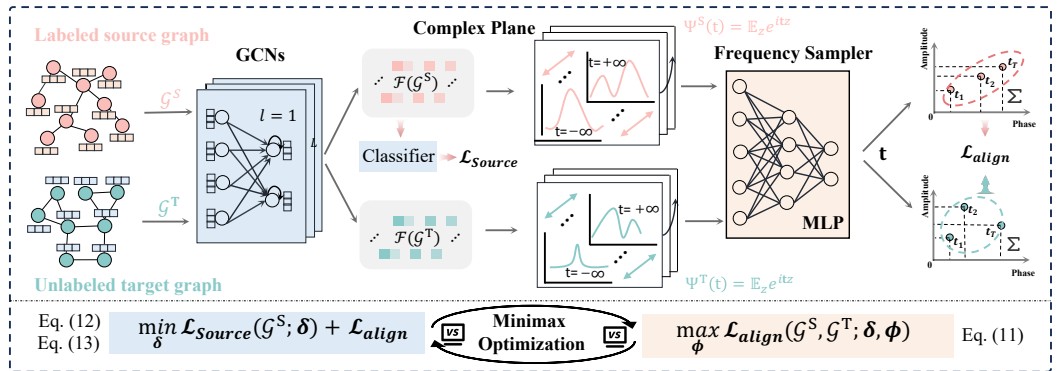

Figure 2: Overall architecture of our proposed ADAlign Framework. The model consists of a source and target branch, each processed by a GNN. A frequency sampler adaptively selects spectral components via a learnable parameter $\phi$, while the classifier is optimized with source labels. The alignment objective $\mathcal{L}_{align}$ is adversarially maximized with respect to $\phi$ and minimized with respect to the model parameters $\delta$, enabling dynamic and efficient distribution matching in the spectral domain.

## 4 METHODOLOGY

In this section, we introduce the detailed workings of ADAlign. As illustrated in Figure 2, we first present the characteristic function transformation in §4.1, followed by NSD in §4.2. In §4.3, we describe the adaptive frequency sampler for NSD, and finally, in §4.4 we introduce the minimax framework that integrates these components for effective graph domain adaptation.

### 4.1 CHARACTERISTIC FUNCTION TRANSFORMATION

ADAlign begins with learning node-level embeddings of the source and target graphs. Formally, we denote the source graph as $\mathcal{G}^{\mathsf{S}} = \{\mathcal{V}^{\mathsf{S}}, \mathcal{E}^{\mathsf{S}}, \mathcal{A}^{\mathsf{S}}, \mathcal{X}^{\mathsf{S}}, \mathcal{Y}^{\mathsf{S}}\}$ and a target graph $\mathcal{G}^{\mathsf{T}} = \{\mathcal{V}^{\mathsf{T}}, \mathcal{E}^{\mathsf{T}}, \mathcal{A}^{\mathsf{T}}, \mathcal{X}^{\mathsf{T}}\}$. We adopt the GNN encoder $\mathcal{F}$ (Liu et al., 2024a) to map each graph into node-level embeddings:

$$Z^{\mathsf{S}} = \mathcal{F}(\mathcal{G}^{\mathsf{S}}), \quad Z^{\mathsf{T}} = \mathcal{F}(\mathcal{G}^{\mathsf{T}}), \tag{3}$$

where $Z^{\mathsf{S}} \in \mathbb{R}^{N_{\mathsf{S}} \times d}$ and $Z^{\mathsf{T}} \in \mathbb{R}^{N_{\mathsf{T}} \times d}$ are the $d$-dimensional node embeddings of nodes in the source and target graphs, respectively. $N_{\mathsf{S}}$ and $N_{\mathsf{T}}$ denote corresponding node numbers. These embeddings serve as compact representations that integrate both attribute and structural information.

Existing approaches (Zhuang et al., 2020; Pang et al., 2023; Yang et al., 2025; Fang et al., 2025b; Chen et al., 2025a) address domain shifts by minimizing statistical distances, e.g., MMD aligns low-order moments while KL divergence matches full densities. Yet, on graph data these criteria often fail: moment matching ignores higher-order dependencies, and density-based measures are unstable in high dimensions. To overcome this, we leverage characteristic function (Waller et al., 1995), which provide complete and tractable representations in the Fourier domain. We extend them to graphs by defining the characteristic functions of the source and target embeddings $Z^{\mathsf{S}}$ and $Z^{\mathsf{T}}$:

$$\Psi^{\mathsf{S}}(\mathbf{t}) = \mathbb{E}_{z \sim \mathrm{P}(Z^{\mathsf{S}})}\left[e^{i\mathbf{t}^{\top}z}\right], \quad \Psi^{\mathsf{T}}(\mathbf{t}) = \mathbb{E}_{z \sim \mathrm{P}(Z^{\mathsf{T}})}\left[e^{i\mathbf{t}^{\top}z}\right], \tag{4}$$

where $\mathrm{P}(Z^{\mathsf{S}})$ and $\mathrm{P}(Z^{\mathsf{T}})$ denote the empirical distributions of node embeddings in the source and target graphs, respectively. Here, $\mathbf{t}$ is a frequency vector in the Fourier domain, $i = \sqrt{-1}$ is the imaginary unit, and $\top$ denotes transposition. The key advantage of CF is that they uniquely characterize probability distributions, offering a complete representation that goes beyond low-order moments or density estimates. The following theorems (Wang et al., 2025) formalize these properties:

**Theorem 1** (Convergence). *(Lévy, 1959; Brézis & Lieb, 1983) Let $\{Z_k\}_{k=1}^{\infty}$ be a sequence of random vectors in $\mathbb{R}^d$ with characteristic functions $\Psi_{Z_k}(\mathbf{t})$. If for every $\mathbf{t} \in \mathbb{R}^d$ the pointwise limit $\lim_{k \to \infty} \Psi_{Z_k}(\mathbf{t}) = \Psi(\mathbf{t})$ exists and $\Psi(\mathbf{t})$ is continuous at the origin, then $\Psi$ is the characteristic function of some random vector $Z$, and we have $Z_k \Rightarrow Z$ as $k \to \infty$.*

**Theorem 2** (Uniqueness). *(Feuerverger & Mureika, 1977; Marcus, 1981) Let $\mathbb{P}$ and $\mathbb{Q}$ be two random vectors with characteristic function $\Psi_{\mathbb{P}}(\mathbf{t})$ and $\Psi_{\mathbb{Q}}(\mathbf{t})$. If $\Psi_{\mathbb{P}}(\mathbf{t}) = \Psi_{\mathbb{Q}}(\mathbf{t}), \quad \forall \mathbf{t} \in \mathbb{R}^d$, then the two random vectors $\mathbb{P}$ and $\mathbb{Q}$ have the same probability distribution.*

## 4.2 Neural Spectral Discrepancy

From the above analysis, probability distributions are uniquely determined by their CFs. This observation suggests that distributional discrepancies in GDA can be naturally assessed in the spectral domain by comparing the CF of the source embeddings $Z^{\mathsf{S}}$ and target embeddings $Z^{\mathsf{T}}$. Motivated by prior works on CF-based discrepancies (Li et al., 2017; 2020; Ansari et al., 2020), we define the *Neural Spectral Discrepancy* (NSD) as a principled metric for cross-graph alignment:

$$\text{NSD}(Z^{\mathsf{S}}, Z^{\mathsf{T}}) = \int_{\mathbf{t} \in \mathcal{T}} \sqrt{\left(\Psi_{Z^{\mathsf{S}}}(\mathbf{t}) - \Psi_{Z^{\mathsf{T}}}(\mathbf{t})\right)\left(\Psi_{Z^{\mathsf{S}}}^*(\mathbf{t}) - \Psi_{Z^{\mathsf{T}}}^*(\mathbf{t})\right)} dF_{\mathcal{T}}(\mathbf{t}), \tag{5}$$

where $\Psi_{Z^{\mathsf{S}}}(\mathbf{t})$ and $\Psi_{Z^{\mathsf{T}}}(\mathbf{t})$ are the CFs of the source and target embeddings, $\Psi^*$ denotes complex conjugation, and $F_{\mathcal{T}}(\mathbf{t})$ is sampling distribution over the frequency domain $\mathcal{T}$. The introduction of complex conjugate ensures the integrand is always real and non-negative (Wang et al., 2025), since

$$\boldsymbol{\ell}(\mathbf{t}) = \left(\Psi_{Z^{\mathsf{S}}}(\mathbf{t}) - \Psi_{Z^{\mathsf{T}}}(\mathbf{t})\right)\left(\Psi_{Z^{\mathsf{S}}}^*(\mathbf{t}) - \Psi_{Z^{\mathsf{T}}}^*(\mathbf{t})\right) = \left|\Psi_{Z^{\mathsf{S}}}(\mathbf{t}) - \Psi_{Z^{\mathsf{T}}}(\mathbf{t})\right|^2. \tag{6}$$

To make this discrepancy more interpretable and amenable to optimization (Ansari et al., 2020), we further exploit the polar decomposition of the complex-valued CF in Eq. (2). This allows $\boldsymbol{\ell}(\mathbf{t})$ to be expressed explicitly in terms of amplitude and phase differences:

$$\boldsymbol{\ell}(\mathbf{t}) = |\Psi_{Z^{\mathsf{S}}}(\mathbf{t})|^2 + |\Psi_{Z^{\mathsf{T}}}(\mathbf{t})|^2 - \Psi_{Z^{\mathsf{S}}}(\mathbf{t})\Psi_{Z^{\mathsf{T}}}^*(\mathbf{t}) - \Psi_{Z^{\mathsf{T}}}(\mathbf{t})\Psi_{Z^{\mathsf{S}}}^*(\mathbf{t})$$

$$= |\Psi_{Z^{\mathsf{S}}}(\mathbf{t})|^2 + |\Psi_{Z^{\mathsf{T}}}(\mathbf{t})|^2 - |\Psi_{Z^{\mathsf{S}}}(\mathbf{t})| |\Psi_{Z^{\mathsf{T}}}(\mathbf{t})|\left(2\cos(\theta_{\mathsf{S}}(\mathbf{t}) - \theta_{\mathsf{T}}(\mathbf{t}))\right)$$

$$= |\Psi_{Z^{\mathsf{S}}}(\mathbf{t})|^2 + |\Psi_{Z^{\mathsf{T}}}(\mathbf{t})|^2 - 2|\Psi_{Z^{\mathsf{S}}}(\mathbf{t})| |\Psi_{Z^{\mathsf{T}}}(\mathbf{t})| + 2|\Psi_{Z^{\mathsf{S}}}(\mathbf{t})| |\Psi_{Z^{\mathsf{T}}}(\mathbf{t})|\left(1 - \cos(\theta_{\mathsf{S}}(\mathbf{t}) - \theta_{\mathsf{T}}(\mathbf{t}))\right)$$

$$= \underbrace{\left(|\Psi_{Z^{\mathsf{S}}}(\mathbf{t})| - |\Psi_{Z^{\mathsf{T}}}(\mathbf{t})|\right)^2}_{\text{amplitude difference}} + 2|\Psi_{Z^{\mathsf{S}}}(\mathbf{t})| |\Psi_{Z^{\mathsf{T}}}(\mathbf{t})| \underbrace{\left(1 - \cos(\theta_{\mathsf{S}}(\mathbf{t}) - \theta_{\mathsf{T}}(\mathbf{t}))\right)}_{\text{phase difference}}, \tag{7}$$

where $\theta_{\mathsf{S}}(\mathbf{t})$ and $\theta_{\mathsf{T}}(\mathbf{t})$ denote the phases of the CF, respectively. This decomposition highlights that NSD naturally separates graph-domain discrepancies into two complementary components: an amplitude term, which captures differences in the concentration of embeddings across frequency components (reflecting global structural variations), and a phase term, which characterizes structural misalignments such as shifts or rotations in relational patterns between graphs. Such a separation provides both interpretability and a more effective basis for optimization. To explicitly balance these two aspects, we consider a convex combination of the two terms via parameter $\kappa$:

$$\boldsymbol{\ell}_{\kappa}(\mathbf{t}) = \kappa\left(|\Psi_{Z^{\mathsf{S}}}(\mathbf{t})| - |\Psi_{Z^{\mathsf{T}}}(\mathbf{t})|\right)^2 + (1-\kappa)\, 2|\Psi_{Z^{\mathsf{S}}}(\mathbf{t})| |\Psi_{Z^{\mathsf{T}}}(\mathbf{t})|\left(1 - \cos(\theta_{\mathsf{S}}(\mathbf{t}) - \theta_{\mathsf{T}}(\mathbf{t}))\right), \tag{8}$$

where $\kappa \in [0, 1]$ controls the trade-off between amplitude discrepancy and phase discrepancy. Building on this discrepancy, we define the alignment loss as:

$$\mathcal{L}_{\text{align}} = \int_{\mathbf{t} \in \mathcal{T}} \sqrt{\boldsymbol{\ell}_{\kappa}(\mathbf{t})}\, dF_{\mathcal{T}}(\mathbf{t}), \tag{9}$$

where $F_{\mathcal{T}}(\mathbf{t})$ denotes the sampling distribution over frequency components. This objective integrates weighted discrepancies across spectral domain, thereby capturing global distributional differences between the source and target graphs while adaptively balancing amplitude and phase shifts.

**Remark.** The amplitude-phase decomposition is motivated by the distinct roles these components play in spectral domain adaptation. Specifically, the amplitude component governs the spectral energy distribution, encapsulating global structural invariants and low-frequency homophilic patterns. In contrast, the phase component encodes the relative positioning and correlation of frequency modes, capturing intricate relational dependencies and heterophilic irregularities. By decoupling these factors, ADAlign facilitates a more granular alignment strategy, allowing for adaptive weighting that yields a more nuanced distribution matching than traditional alignment approaches.

## 4.3 Adaptive Frequency Sampler for NSD

A central challenge in spectral alignment is the choice of frequency points $\mathbf{t}$ at which the characteristic functions are evaluated. Effective alignment of amplitude and phase discrepancies requires sampling informative frequencies $\mathbf{t}$. However, fixed or heuristic grids are either too sparse (missing subtle but critical shifts) or too dense (incurring redundant computation and noise). We therefore learn the frequency sampling distribution by parameterizing $F_{\mathcal{T}}$ with a density $p_{\mathcal{T}}(\mathbf{t}; \phi)$ over the spectral domain $\mathcal{T}$. Concretely, we model $p_{\mathcal{T}}(\mathbf{t}; \phi)$ as a normal scale mixture (Ansari et al., 2020; Li et al., 2023; Wang et al., 2025), which affords both flexibility and analytical convenience:

$$p_{\mathcal{T}}(\mathbf{t}; \phi) = \int_{\Sigma} p_N(\mathbf{t} \mid \mathbf{0}, \Sigma) \, p_{\Sigma}(\Sigma; \phi) \, d\Sigma, \tag{10}$$

where $p_N(\mathbf{t} \mid \mathbf{0}, \Sigma)$ is a Gaussian with covariance $\Sigma$, and $p_{\Sigma}(\Sigma; \phi)$ is a neural mixing distribution. Normal scale mixtures subsume common families (e.g., Gaussian, Cauchy, Student-$t$), enabling adaptive emphasis on low frequencies (global variations) or high frequencies (fine-grained shifts). We use the reparameterization trick to ensure that sampling $\mathbf{t}$ remains differentiable. The sampler is trained to prioritize frequencies that expose larger spectral discrepancies. Denoting the pointwise NSD contribution by $\sqrt{\ell_{\kappa}(\mathbf{t})}$ ( Eq. (8)), we maximize its expectation under $p_{\mathcal{T}}$:

$$\max_{\phi} \ \mathcal{L}_{\text{align}}(\phi) = \int_{\mathbf{t} \in \mathcal{T}} \sqrt{\ell_{\kappa}(\mathbf{t})} \, p_{\mathcal{T}}(\mathbf{t}; \phi) \, d\mathbf{t} \ \approx \ \frac{1}{M} \sum_{m=1}^{M} \sqrt{\ell_{\kappa}(\mathbf{t}_m)}, \quad \mathbf{t}_m \sim p_{\mathcal{T}}(\cdot; \phi), \tag{11}$$

where $M$ is the number of sampled frequencies. As $M$ increases, the Monte Carlo approximation converges to the true expectation, and by Theorem 1 the empirical CF becomes more accurate, ultimately leading to higher-quality distributional alignment. This objective shifts probability mass toward frequency regions that most contribute to source-target discrepancy, yielding an adaptive spectral sampler for NSD and a compact, high-signal set of frequencies for downstream alignment.

## 4.4 Minimax Optimization Framework

To achieve effective domain adaptation, ADAlign jointly optimizes two complementary objectives: (i) preserving task-specific discriminability on the source domain, and (ii) reducing cross-domain distributional gaps through spectral alignment. Let $\delta$ denote the learnable parameters of the GNN. The source objective is the standard cross-entropy loss on labeled source nodes:

$$\min_{\delta} \ \mathcal{L}_{\text{source}} = \min_{\delta} \ \text{CE}\big(\text{GNN}_{\delta}(\mathcal{X}^{\mathsf{S}}, \mathcal{A}^{\mathsf{S}}), \mathcal{Y}^{\mathsf{S}}\big), \tag{12}$$

where $(\mathcal{X}^{\mathsf{S}}, \mathcal{A}^{\mathsf{S}}, \mathcal{Y}^{\mathsf{S}})$ are the source node features, adjacency matrix, and labels, and $\text{CE}(\cdot)$ denotes cross-entropy. In parallel, we align the source and target graph representations in the spectral domain. Let $\phi$ be the parameters of frequency sampler (Eq. (10)), then the alignment objective is:

$$\min_{\delta} \ \mathcal{L}_{\text{align}} = \int_{\mathbf{t} \in \mathcal{T}} \sqrt{\ell_{\kappa}(\mathbf{t})} \, p_{\mathcal{T}}(\mathbf{t}; \phi) \, d\mathbf{t}. \tag{13}$$

Combining Eq. (11), Eq. (12) and Eq. (13), we obtain a minimax formulation:

$$\min_{\delta} \ \max_{\phi} \ \Big[ \mathcal{L}_{\text{source}}(\mathcal{G}^{\mathsf{S}}; \delta) + \lambda \, \mathcal{L}_{\text{align}}(\mathcal{G}^{\mathsf{S}}, \mathcal{G}^{\mathsf{T}}; \delta, \phi) \Big], \tag{14}$$

where $\lambda$ balances the two objectives. In this framework, the GNN parameters $\delta$ are optimized to jointly minimize the classification and alignment losses, while the sampler parameters $\phi$ are updated adversarially to emphasize spectral regions with large domain discrepancies.

Due to space limitations, the training procedure and complexity analysis are provided in Appendix C.

## 4.5 Theoretical Analysis

In this section, we provide a PAC-Bayesian analysis for proposed ADAlign framework (Ma et al., 2021; Fang et al., 2025b). To evaluate the generalization performance of a deterministic GDA classifier on structured graph data, we begin by introducing the notion of margin loss on a graph.

**Margin Loss on a Graph.** Consider a graph $\mathcal{G} = (\mathcal{V}, \mathcal{E}, \mathcal{A}, \mathcal{X}, \mathcal{Y})$, where $\mathcal{V}$ and $\mathcal{E}$ denote the nodes and edges, $\mathcal{A}$ is the adjacency matrix, $\mathcal{X}$ is the feature matrix, and $\mathcal{Y}$ the node labels. For a classifier $h$ and margin $\gamma \geq 0$, the empirical margin loss is defined as $\tilde{\mathcal{L}}_\gamma(h) :=$ $\frac{1}{|\mathcal{V}|} \sum_{i \in \mathcal{V}} \mathbf{1}[h_i(\mathcal{X}, \mathcal{G})[\mathcal{Y}_i] \leq \gamma + \max_{c \neq \mathcal{Y}_i} h_i(\mathcal{X}, \mathcal{G})[c]]$, where $\mathbf{1}[\cdot]$ is the indicator function and $c$ ranges over possible labels. The expected margin loss is then $\mathcal{L}_\gamma(h) := \mathbb{E}_{i \in \mathcal{V}, Y_i \sim \mathrm{P}(\mathcal{Y}|Z_i)} \tilde{\mathcal{L}}_\gamma(h)$, with $Y_i \sim \mathrm{P}(\mathcal{Y} \mid Z_i)$ denoting the conditional label distribution given node embedding $Z_i$.

**Theorem 3** (GDA Bound for Deterministic Classifiers). *Let $\mathcal{H}$ be a hypothesis class. For any $h \in \mathcal{H}$, $\xi > 0$, and $\gamma \geq 0$, let $\mathbb{P}$ be a prior distribution on $\mathcal{H}$ that is independent of the source sample $\mathcal{V}^{\mathsf{S}}$. Then, with probability at least $1 - \rho$ over $\mathcal{Y}^{\mathsf{S}}$, for any posterior $\mathbb{Q}$ on $\mathcal{H}$ it holds that*

$$\mathcal{L}_0^{\mathsf{T}}(h) \leq \tilde{\mathcal{L}}_\gamma^{\mathsf{S}}(h) + \frac{1}{\xi}\left[2\big(\mathsf{D}_{\mathrm{KL}}(\mathbb{P} \parallel \mathbb{Q}) + 1\big) + \ln\frac{1}{\rho} + \frac{\xi^2}{4N_S} + \mathsf{D}_{\gamma/2}^{\mathsf{S},\mathsf{T}}(\mathbb{P}; \xi)\right]. \tag{15}$$

In this bound, $\mathsf{D}_{\mathrm{KL}}(\mathbb{P} \parallel \mathbb{Q})$ denotes the Kullback-Leibler divergence between distributions $\mathbb{P}$ and $\mathbb{Q}$, which serves as a model complexity measure. The terms $\ln\frac{1}{\rho}$ and $\frac{\xi^2}{4N_S}$ are standard in PAC-Bayesian bounds, while the $\mathsf{D}_{\gamma/2}^{\mathsf{S},\mathsf{T}}(\mathbb{P}; \xi)$ captures the difference between source nodes $\mathcal{V}^{\mathsf{S}}$ and target nodes $\mathcal{V}^{\mathsf{T}}$.

**Proposition 1** $\left(\text{Bound for } \mathsf{D}_{\gamma/2}^{\mathsf{S},\mathsf{T}}(\mathbb{P}; \xi)\right)$. For any $\gamma \geq 0$, if the prior $\mathbb{P}$ over $\mathcal{H}$ is fixed, the domain discrepancy $\mathsf{D}_{\gamma/2}^{\mathsf{S},\mathsf{T}}(\mathbb{P}; \xi)$ can be bounded by a weighted gap in the spectral domain:

$$\mathsf{D}_{\gamma/2}^{\mathsf{S},\mathsf{T}}(\mathbb{P}; \xi) \leq \mathcal{O}\left(\sum_{m=1}^{M} \omega_m \left|\Psi^{\mathsf{S}}(\mathbf{t}_m) - \Psi^{\mathsf{T}}(\mathbf{t}_m)\right|^2\right), \tag{16}$$

where $\{\mathbf{t}_m\}_{m=1}^M \subset \mathbb{R}^d$ are sampled frequencies, and $\{\omega_m\}_{m=1}^M$ are nonnegative weights approximating a spectral measure. This result shows that domain discrepancy can be upper-bounded by the weighted frequency-domain gap between source and target CFs, with $\omega_m$ reflecting the importance of each spectral component. Crucially, learning these weights adaptively enables the model to emphasize discriminative frequencies that drive cross-domain differences. Our ADAlign framework is directly aligned with this principle: by dynamically prioritizing informative frequencies, it achieves tighter bounds and more effective graph domain adaptation. Full proofs are deferred to Appendix B.

## 5 EXPERIMENTS

In this section, we empirically evaluate the ADAlign framework through experiments addressing five key research questions: **RQ1**: How does ADAlign compare to state-of-the-art methods? **RQ2**: What are the computational efficiency and resource overhead of ADAlign? **RQ3**: Does the adaptive frequency sampler effectively capture meaningful spectral components? **RQ4**: How do key parameters impact model performance? **RQ5**: How can we intuitively understand ADAlign's strengths?

### 5.1 EXPERIMENTAL SETUP

**(1) Datasets.** To demonstrate the superiority of proposed ADAlign framework, we evaluate it across four types of datasets, comprising a total of ten datasets (Fang et al., 2025a): ArnetMiner (ACMv9, Citationv1, DBLPv7) (Dai et al., 2022), Airport (USA, Brazil, Europe) (Ribeiro et al., 2017), Blog (Blog1, Blog2) (Li et al., 2015), and Twitch (England, Germany) (Liu et al., 2024a). **(2) Baselines.** To assess the effectiveness of our model, we conduct comparisons with two categories of baseline methods: (i) source-only graph neural networks such as GAT, GIN, and GCN, which are exclusively trained on the source domain before being applied to the target domain; and (ii) graph domain adaptation techniques, including DANE, UDAGCN, AdaGCN, StruRW, SA-GDA, GRADE, PairAlign, GraphAlign, A2GNN, TDSS, DGSDA, GAA and HGDA. **(3) Implementation Details.** We utilize Micro-F1 and Macro-F1 scores as evaluation metrics for our experiments. More dataset descriptions, baselines, and implementation details can be found in Appendix D.

### 5.2 MAIN EXPERIMENTAL RESULTS (**RQ1**)

To assess the effectiveness of ADAlign framework, we conduct comprehensive evaluations on ten benchmark datasets covering four diverse graph domains, with results summarized in Table 1.

| Methods | A→C | A→D | C→A | C→D | D→A | D→C | B1→B2 | B2→B1 |
|---|---|---|---|---|---|---|---|---|
| GAT (ICLR'18) | $62.77_{\pm2.24}$ | $60.29_{\pm1.33}$ | $58.35_{\pm1.51}$ | $67.07_{\pm2.05}$ | $54.33_{\pm2.17}$ | $63.24_{\pm2.21}$ | $21.15_{\pm2.72}$ | $19.46_{\pm1.41}$ |
| GIN (ICLR'19) | $69.95_{\pm0.90}$ | $64.69_{\pm1.43}$ | $62.63_{\pm0.23}$ | $68.54_{\pm0.31}$ | $58.18_{\pm0.60}$ | $69.91_{\pm1.83}$ | $20.51_{\pm2.16}$ | $18.60_{\pm0.38}$ |
| GCN (ICLR'17) | $70.82_{\pm1.26}$ | $65.05_{\pm2.15}$ | $65.44_{\pm1.14}$ | $69.46_{\pm0.83}$ | $59.92_{\pm0.72}$ | $66.83_{\pm0.94}$ | $23.17_{\pm1.10}$ | $23.46_{\pm0.48}$ |
| DANE (IJCAI'19) | $69.77_{\pm2.14}$ | $62.41_{\pm2.15}$ | $63.93_{\pm1.32}$ | $65.05_{\pm2.07}$ | $58.21_{\pm1.17}$ | $67.41_{\pm2.29}$ | $28.15_{\pm1.77}$ | $30.32_{\pm0.69}$ |
| UDAGCN (WWW'20) | $80.68_{\pm0.31}$ | $74.66_{\pm0.93}$ | $73.46_{\pm0.40}$ | $76.97_{\pm0.31}$ | $69.36_{\pm0.31}$ | $77.81_{\pm0.50}$ | $33.87_{\pm2.09}$ | $31.86_{\pm1.08}$ |
| AdaGCN (TKDE'22) | $68.07_{\pm0.86}$ | $66.72_{\pm1.07}$ | $63.25_{\pm1.15}$ | $70.97_{\pm0.48}$ | $60.68_{\pm0.67}$ | $65.83_{\pm2.33}$ | $30.75_{\pm0.64}$ | $27.17_{\pm2.34}$ |
| StruRW (ICML'23) | $60.48_{\pm1.41}$ | $59.63_{\pm3.04}$ | $56.18_{\pm0.86}$ | $62.50_{\pm1.40}$ | $52.25_{\pm1.15}$ | $57.55_{\pm0.84}$ | $40.37_{\pm0.44}$ | $42.01_{\pm0.44}$ |
| SA-GDA (MM'23) | $76.02_{\pm1.57}$ | $70.36_{\pm1.08}$ | $70.84_{\pm1.87}$ | $75.48_{\pm0.20}$ | $63.41_{\pm1.12}$ | $72.81_{\pm1.66}$ | $49.36_{\pm1.83}$ | $45.57_{\pm2.33}$ |
| GRADE (AAAI'23) | $75.02_{\pm0.48}$ | $68.17_{\pm0.25}$ | $68.96_{\pm0.10}$ | $73.48_{\pm0.30}$ | $61.72_{\pm0.36}$ | $71.69_{\pm0.90}$ | $48.44_{\pm3.12}$ | $\underline{46.78}_{\pm1.31}$ |
| PairAlign (ICML'24) | $60.25_{\pm0.89}$ | $59.58_{\pm0.58}$ | $56.02_{\pm0.85}$ | $63.49_{\pm0.70}$ | $51.83_{\pm0.69}$ | $58.60_{\pm0.42}$ | $40.01_{\pm0.78}$ | $42.64_{\pm0.69}$ |
| GraphAlign (KDD'24) | $75.18_{\pm0.62}$ | $68.81_{\pm0.78}$ | $65.21_{\pm0.55}$ | $72.21_{\pm0.45}$ | $61.66_{\pm1.23}$ | $68.85_{\pm0.56}$ | $45.58_{\pm1.15}$ | $43.21_{\pm0.62}$ |
| A2GNN (AAAI'24) | $\underline{80.93}_{\pm0.52}$ | $\underline{75.94}_{\pm0.33}$ | $\underline{75.09}_{\pm0.43}$ | $77.16_{\pm0.23}$ | $\underline{73.21}_{\pm0.44}$ | $\underline{79.72}_{\pm0.63}$ | $47.10_{\pm3.40}$ | $44.01_{\pm2.01}$ |
| TDSS (AAAI'25) | $80.41_{\pm0.71}$ | $74.04_{\pm3.64}$ | $72.88_{\pm2.83}$ | $\underline{77.23}_{\pm2.71}$ | $72.38_{\pm1.95}$ | $79.04_{\pm3.61}$ | $\underline{49.53}_{\pm1.08}$ | $44.20_{\pm0.29}$ |
| GAA (ICLR'25) | $80.03_{\pm0.37}$ | $73.32_{\pm0.67}$ | $73.15_{\pm0.32}$ | $76.04_{\pm0.39}$ | $68.32_{\pm0.43}$ | $78.27_{\pm0.31}$ | $48.80_{\pm2.45}$ | $43.74_{\pm1.25}$ |
| DGSDA (ICML'25) | $78.59_{\pm0.55}$ | $73.71_{\pm0.24}$ | $73.96_{\pm0.70}$ | $76.56_{\pm0.25}$ | $72.69_{\pm0.34}$ | $77.41_{\pm0.11}$ | $48.61_{\pm1.73}$ | $44.12_{\pm1.26}$ |
| HGDA (ICML'25) | $76.61_{\pm1.03}$ | $72.18_{\pm3.91}$ | $67.11_{\pm0.93}$ | $73.63_{\pm0.79}$ | $66.52_{\pm0.36}$ | $75.23_{\pm1.03}$ | $46.34_{\pm1.93}$ | $44.29_{\pm1.60}$ |
| **ADAlign (Ours)** | $\mathbf{82.21}_{\pm0.36}$ | $\mathbf{78.06}_{\pm0.61}$ | $\mathbf{76.24}_{\pm0.29}$ | $\mathbf{79.51}_{\pm0.20}$ | $\mathbf{74.38}_{\pm0.98}$ | $\mathbf{81.84}_{\pm0.10}$ | $\mathbf{55.08}_{\pm0.25}$ | $\mathbf{52.42}_{\pm0.46}$ |
| Methods | U→B | U→E | B→U | B→E | E→U | E→B | DE→EN | EN→DE |
| GAT (ICLR'18) | $50.53_{\pm3.25}$ | $40.25_{\pm3.82}$ | $44.25_{\pm1.30}$ | $46.67_{\pm1.99}$ | $43.90_{\pm1.69}$ | $50.84_{\pm3.22}$ | $57.65_{\pm0.14}$ | $63.13_{\pm0.31}$ |
| GIN (ICLR'19) | $33.13_{\pm2.96}$ | $32.28_{\pm4.55}$ | $35.87_{\pm1.96}$ | $32.98_{\pm2.54}$ | $36.07_{\pm2.44}$ | $33.89_{\pm4.03}$ | $55.10_{\pm0.56}$ | $60.53_{\pm0.04}$ |
| GCN (ICLR'17) | $56.34_{\pm3.75}$ | $47.11_{\pm0.52}$ | $40.22_{\pm1.21}$ | $48.92_{\pm1.88}$ | $44.67_{\pm0.99}$ | $58.47_{\pm2.39}$ | $57.59_{\pm0.26}$ | $62.82_{\pm0.33}$ |
| DANE (IJCAI'19) | $44.12_{\pm1.56}$ | $39.90_{\pm3.25}$ | $42.20_{\pm4.43}$ | $38.45_{\pm3.29}$ | $36.13_{\pm7.25}$ | $46.56_{\pm0.48}$ | $55.44_{\pm1.12}$ | $62.00_{\pm0.90}$ |
| UDAGCN (WWW'20) | $58.17_{\pm2.07}$ | $44.51_{\pm0.41}$ | $42.17_{\pm1.14}$ | $47.97_{\pm1.72}$ | $42.79_{\pm1.47}$ | $62.29_{\pm3.79}$ | $59.37_{\pm0.63}$ | $63.61_{\pm0.17}$ |
| AdaGCN (TKDE'22) | $65.65_{\pm1.93}$ | $50.63_{\pm0.55}$ | $46.87_{\pm1.00}$ | $51.44_{\pm0.93}$ | $48.62_{\pm0.53}$ | $\underline{73.74}_{\pm2.24}$ | $57.81_{\pm0.33}$ | $62.67_{\pm0.39}$ |
| StruRW (ICML'23) | $62.44_{\pm1.96}$ | $39.70_{\pm1.71}$ | $41.45_{\pm1.30}$ | $38.30_{\pm1.52}$ | $41.21_{\pm1.50}$ | $55.11_{\pm0.75}$ | $56.72_{\pm1.09}$ | $61.01_{\pm0.09}$ |
| SA-GDA (MM'23) | $67.60_{\pm1.54}$ | $51.09_{\pm1.02}$ | $48.20_{\pm3.29}$ | $\underline{56.99}_{\pm0.32}$ | $47.78_{\pm2.42}$ | $63.63_{\pm1.82}$ | $56.46_{\pm1.39}$ | $62.18_{\pm0.34}$ |
| GRADE (AAAI'23) | $57.71_{\pm0.78}$ | $48.97_{\pm0.20}$ | $43.31_{\pm0.58}$ | $55.94_{\pm0.19}$ | $46.64_{\pm0.80}$ | $63.05_{\pm0.97}$ | $59.16_{\pm0.24}$ | $64.11_{\pm0.03}$ |
| PairAlign (ICML'24) | $66.26_{\pm1.77}$ | $39.50_{\pm0.95}$ | $41.92_{\pm2.30}$ | $38.10_{\pm2.60}$ | $39.63_{\pm0.94}$ | $55.57_{\pm0.75}$ | $57.45_{\pm0.12}$ | $\underline{64.22}_{\pm0.18}$ |
| GraphAlign (KDD'24) | $62.54_{\pm0.80}$ | $\underline{52.18}_{\pm0.17}$ | $50.33_{\pm0.77}$ | $55.23_{\pm2.33}$ | $\underline{54.35}_{\pm0.33}$ | $71.02_{\pm0.48}$ | $52.58_{\pm0.07}$ | $60.12_{\pm0.01}$ |
| A2GNN (AAAI'24) | $64.24_{\pm1.37}$ | $46.62_{\pm1.61}$ | $47.66_{\pm5.51}$ | $47.67_{\pm1.35}$ | $52.72_{\pm1.16}$ | $54.81_{\pm1.63}$ | $55.77_{\pm0.47}$ | $60.92_{\pm0.56}$ |
| TDSS (AAAI'25) | $67.43_{\pm1.62}$ | $52.05_{\pm1.01}$ | $47.05_{\pm2.07}$ | $51.80_{\pm0.12}$ | $46.08_{\pm0.51}$ | $55.73_{\pm1.08}$ | $55.91_{\pm0.47}$ | $61.09_{\pm0.32}$ |
| GAA (ICLR'25) | $64.89_{\pm2.38}$ | $51.70_{\pm1.97}$ | $52.50_{\pm1.67}$ | $48.22_{\pm1.07}$ | $48.70_{\pm2.34}$ | $56.07_{\pm1.51}$ | $57.54_{\pm0.37}$ | $61.98_{\pm0.40}$ |
| DGSDA (ICML'25) | $61.22_{\pm1.63}$ | $49.35_{\pm2.78}$ | $\underline{54.91}_{\pm0.81}$ | $42.56_{\pm1.64}$ | $49.76_{\pm0.82}$ | $60.15_{\pm0.57}$ | $\underline{60.84}_{\pm0.15}$ | $63.42_{\pm0.30}$ |
| HGDA (ICML'25) | $67.82_{\pm1.30}$ | $48.67_{\pm2.11}$ | $51.94_{\pm2.66}$ | $48.07_{\pm1.94}$ | $46.62_{\pm1.89}$ | $59.77_{\pm0.89}$ | $57.08_{\pm0.36}$ | $61.19_{\pm0.91}$ |
| **ADAlign (Ours)** | $\mathbf{72.52}_{\pm0.41}$ | $\mathbf{53.23}_{\pm0.68}$ | $\mathbf{55.71}_{\pm0.59}$ | $\mathbf{58.49}_{\pm1.19}$ | $\mathbf{55.41}_{\pm0.29}$ | $\mathbf{75.82}_{\pm0.47}$ | $\mathbf{61.92}_{\pm0.12}$ | $\mathbf{65.69}_{\pm0.26}$ |

Table 1: Node classification performance ($\% \pm \sigma$, Micro-F1 score) under 16 transfer scenarios. The highest scores are highlighted in **bold**, while the second-highest scores are underlined.

| Model/Datasets | A→C | | A→D | |
|---|---|---|---|---|
| | Speed (s) | Memory (GB) | Speed (s) | Memory (GB) |
| A2GNN (AAAI'24) | 0.1812 | 10.64 | 0.1781 | 10.45 |
| TDSS (AAAI'25) | 0.1684 | 10.64 | 0.1661 | 10.51 |
| GAA (ICLR'25) | 0.2662 | 15.80 | 0.2520 | 15.60 |
| HGDA (ICML'25) | 0.3715 | 22.31 | 0.3817 | 21.07 |
| **ADAlign (Ours)** | **0.08 (2.0× ↑)** | **3.45 (3.1× ↓)** | **0.07 (2.3× ↑)** | **2.60 (4.0× ↓)** |

Table 2: Per-iteration runtime and memory consumption comparison on two transfer tasks. Values in parentheses show relative improvement: "↑" speedup, "↓" memory reduction.

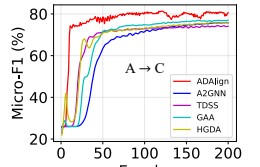
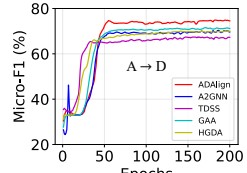

Figure 4: The comparative training curves of Micro-F1 scores are illustrated for four representative baselines across two distinct transfer scenarios over 200 epochs of iterative learning.

ADAlign consistently achieves the best performance across all 16 transfer scenarios, demonstrating its ability to capture and align complex composite distribution shifts. These improvements are attributed to ADAlign's ability to adaptively capture multi-level distribution shifts through NSD modeling, rather than relying on fixed heuristic factors such as GAA and HGDA. In particular, ADAlign yields an average improvement of over 6% in challenging heterogeneous blog-domain transfers, where the extremely high-dimensional features and multiple label categories create more entangled shifts, highlighting its robustness under severe distribution shifts. These results confirm the effectiveness and generality of holistic spectral alignment for graph domain adaptation. Moreover, the consistent gains across academic, social, and airport graphs demonstrate that ADAlign is not tied to a specific domain, but rather provides a general-purpose solution. Additional experimental results and significance tests ($p < 0.05$) are provided in Appendix E.1 and Appendix E.2.

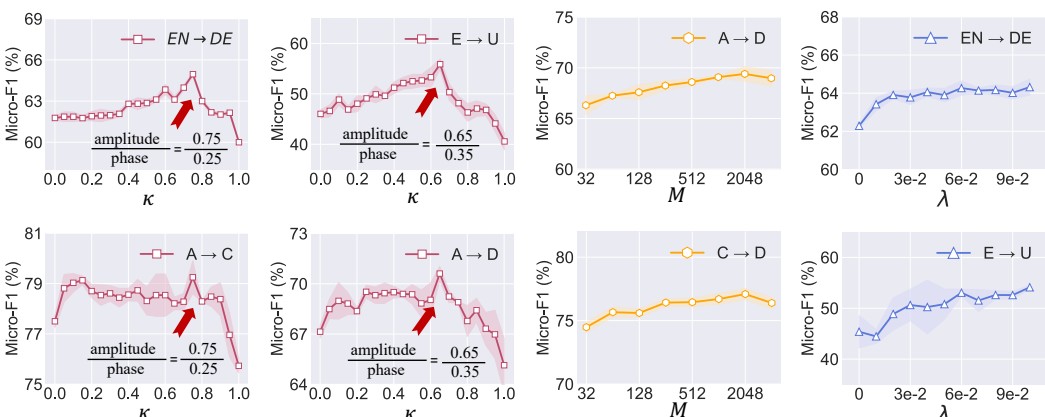

Figure 6: The performances of our ADAlign w.r.t varying parameters on different transfer tasks.

## 5.3 ANALYSIS ON EFFICIENCY AND STABILITY (**RQ2**)

We compare runtime, memory, and training dynamics against competitive baselines. As shown in Table 2, ADAlign effectively reduces runtime and memory usage. On the A→C task, for instance, it achieves 0.08 s/iter with 3.45 GB memory, representing a 2.0× speedup and a 3.1× reduction in memory usage compared to TDSS, the strongest competitor in efficiency. This advantage arises from using a more efficient metric that avoids costly kernel computations in traditional methods (e.g., MMD). By leveraging neural characteristic function, we avoid the computational burden of large-scale kernel matrices and the high variance introduced by sampling-based estimations, leading to improvements in both runtime and memory efficiency while preserving the flexibility needed for graph domain adaptation. Furthermore, Figure 4 illustrates that ADAlign trains efficiently and stably. In contrast to adversarial methods that often suffer from instability, ADAlign remains smooth and robust. This combination of low computational cost and stable training dynamics makes ADAlign practical for real-world deployment. The training loss curves are provided in Appendix E.4.

## 5.4 ABLATION STUDIES ON FREQUENCY SAMPLER (**RQ3**)

We ablate the adaptive frequency (AF) sampler by comparing ADAlign with three variants: Random Frequencies (RF), Low Frequencies (LF), and High Frequencies (HF). For LF, we select the frequency range [1, 10], and for HF, the range [10, 20]. These ranges are determined based on the similarity between transfer scenarios, with the highest frequency dynamically scaled according to a similarity measure

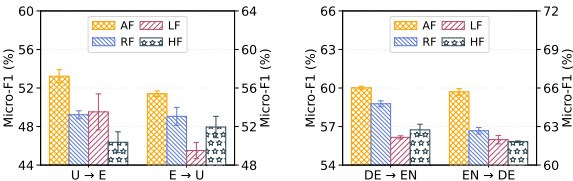

Figure 5: Classification Micro-F1 comparisons between ADA variants on four cross-domain tasks.

$S$ between the source and target graphs. Specifically, the highest frequency $f_{\max}$ is given by: $f_{\max} = f_{\text{base}} \times (1 - S/S_{\max})$, where $f_{\text{base}}$ is the base frequency, $S$ is the similarity measure between the graphs, and $S_{\max}$ is the maximum similarity. This scaling ensures that the frequency range adapts to the graph's structure and the level of discrepancy between the domains. As shown in Figure 5, AF consistently achieves the best performance across four transfer tasks, confirming that dominant shifts lie in scenario-specific spectral components. The task-level patterns are intuitive: LF performs competitively on U→E and E→U, reflecting the role of smooth, large-scale structures, while HF underperforms on DE→EN and EN→DE, indicating that ignoring fine-grained cues is detrimental. RF yields unstable and suboptimal results, suggesting that unguided sampling fails to target critical discrepancies. Overall, these findings underscore the benefit of our minimax formulation: instead of relying on a fixed heuristic band, it adaptively identifies and emphasizes the most discriminative frequencies, enabling precise alignment. Additional ablation results are provided in Appendix E.5.

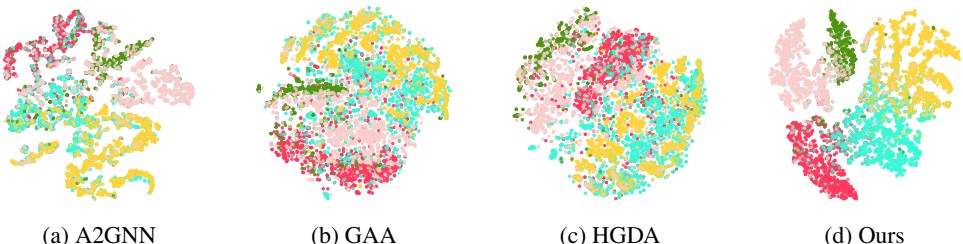

| (a) A2GNN | (b) GAA | (c) HGDA | (d) Ours |

Figure 7: Visualization of node embeddings inferred by four models for the A → C task.

## 5.5 PARAMETER SENSITIVITY ANALYSIS (**RQ4**)

In this section, we analyze the effect of three key hyperparameters in ADAlign: the amplitude–phase trade-off $\kappa$ in Eq. (8), the sampled-frequency number $M$, and the alignment loss weight $\lambda$ in Eq. (14). The corresponding results are shown in Figure 6. Additional results are provided in Appendix E.6.

**Study on amplitude weight $\kappa$.** The parameter $\kappa$ controls the relative emphasis on amplitude versus phase in the spectral domain. We find that performance is consistently strong when $\kappa$ lies in the range $0.65$ and $0.75$, highlighting the importance of balancing global structural consistency (amplitude) with fine-grained relational cues (phase). Worth noting is that Pushing $\kappa$ toward either extreme ($\kappa = 1.0$ or $\kappa = 0.0$) leads to clear degradation, confirming the necessity of joint alignment.

**Study on the number of frequencies $M$.** The parameter $M$ determines the number of sampled frequency components used to compute the discrepancy. Increasing $M$ improves performance at first, since richer frequency coverage captures more detailed aspects of distributional shift. However, gains plateau beyond $M = 2048$, suggesting that ADAlign can effectively capture essential spectral characteristics without requiring excessive samples, thereby retaining computational efficiency.

**Study on loss weight $\lambda$.** The hyperparameter $\lambda$ balances the domain alignment loss against the source classification loss. As $\lambda$ increases, performance improves steadily before saturating, indicating that moderate alignment weighting suffices to achieve strong transfer. Very small $\lambda$ under-emphasizes alignment, while overly large $\lambda$ slightly impairs source discrimination. Overall, ADAlign remains robust, with mid-range values of $\lambda$ consistently providing an effective trade-off.

## 5.6 VISUALIZATION OF NODE EMBEDDINGS (**RQ5**)

To intuitively demonstrate the superiority of ADAlign, we visualize the node embeddings learned by different methods for the A → C task using t-SNE (Maaten & Hinton, 2008), as illustrated in Figure 7. A contrast can be observed across the visualizations. The embeddings generated by baseline models, such as A2GNN, GAA, and HGDA, show significant overlap and intermingling between classes from the source and target domains. This indicates that these methods struggle to learn representations that are both domain-invariant and class-discriminative, leading to suboptimal adaptation performance. In contrast, the embeddings produced by our framework demonstrate two desirable characteristics. First, samples from the same class across both domains form compact, well-aligned clusters, highlighting the effectiveness of our spectral alignment mechanism in learning domain-invariant features. Second, the boundaries between different classes are sharp and well-separated, showing that our minimax alignment over characteristic function not only mitigates domain shift but also preserves task-relevant discriminative information.

## 6 CONCLUSION

In this paper, we introduced the ADAlign framework, a novel approach to handling composite distribution shifts in GDA. By leveraging characteristic function in the complex Fourier domain, ADAlign dynamically identifies and aligns discriminative spectral components through minimax optimization, eliminating the need for manual reweighting or heuristic metrics. Extensive experiments across 16 benchmarks show that ADAlign consistently outperforms state-of-the-art methods, while also improving training stability and reducing computational overhead. These advantages make ADAlign a highly effective and practical solution for real-world graph transfer learning tasks.

## ACKNOWLEDGEMENTS

This work was supported by the National Key Research and Development Program of China under Grant Nos. 2024YFF0729003, the National Natural Science Foundation of China under Grant Nos. 62176014, 62276015, 62206266, the Fundamental Research Funds for the Central Universities.

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

# Appendix

## Table of Contents

## A  THE USE OF LARGE LANGUAGE MODELS (LLMS)

In this work, we utilized LLMs solely for the purpose of English language refinement. These models were employed to assist with the proofreading and enhancement of written text, ensuring clarity, coherence, and grammatical accuracy. The LLMs were not used for generating content, and all research, analysis, and conclusions presented are the result of our own work and independent thought.

## B  PROOF OF PROPOSITION 1

**Definition 1.** *The nodes are partitioned into two disjoint classes, denoted as $c_0$ and $c_1$. Each node feature vector $\mathbf{x}$ is sampled from a Gaussian distribution $N(\boldsymbol{\mu}_i, \sigma_i)$ for $i \in \{0, 1\}$. The source and target graphs each contain nodes from both classes, denoted as $c_i^{(\mathsf{S})}$ and $c_i^{(\mathsf{T})}$, respectively. The class distribution is balanced, i.e., $P(Y = c_0) = P(Y = c_1)$.*

**Theorem 4.** *For any two nodes $s \in \mathcal{V}^{\mathsf{S}}$ and $t \in \mathcal{V}^{\mathsf{T}}$ with aggregated feature representations $Z = GNN(\mathbf{x})$, the following inequality (Mao et al., 2023; Fang et al., 2025a) holds:*

$$\|P(y_u = c_0 \mid Z_u) - P(y_v = c_0 \mid Z_v)\| \leq \mathcal{O}\left(\|Z_u - Z_v\| + \|Z_v - \boldsymbol{\mu}_1^{(\mathsf{S})}\| + \|Z_v - \boldsymbol{\mu}_1^{(\mathsf{T})}\|\right). \quad (17)$$

**Proposition 1** $\left(\text{Bound for } \mathsf{D}_{\gamma/2}^{\mathsf{S},\mathsf{T}}(\mathbb{P}; \xi)\right)$**.** *For any $\gamma \geq 0$, under the assumption that the prior distribution $\mathbb{P}$ over the function family $\mathcal{H}$ is specified, the domain discrepancy $\mathsf{D}_{\gamma/2}^{\mathsf{S},\mathsf{T}}(\mathbb{P}; \xi)$ can be bounded as follows:*

$$\mathsf{D}_{\gamma/2}^{\mathsf{S},\mathsf{T}}(\mathbb{P}; \xi) \leq \mathcal{O}\left(\sum_{m=1}^{M} \omega_m \left|\Psi^{\mathsf{S}}(\mathbf{t}_m) - \Psi^{\mathsf{T}}(\mathbf{t}_m)\right|^2\right), \quad (18)$$

*where $\{\mathbf{t}_m\}_{m=1}^{M} \subset \mathbb{R}^d$ are frequency samples, and $\{\omega_m\}_{m=1}^{M}$ are nonnegative quadrature weights that approximate a target spectral measure.*

*Proof.* For notational simplicity, let $h_i \equiv h_i(\mathcal{X}, \mathcal{G})$ for any $i \in \mathcal{V}^\mathsf{S} \cup \mathcal{V}^\mathsf{T}$. Define $\eta_k(i) = \mathbb{P}(y_i = k \mid Z_i)$ for $k \in \{0, 1\}$, and let $\mathcal{L}_\gamma(h_i, y_i) = \mathbb{1}\left[h_i[y_i] \leq \gamma + \max_{k \neq y_i} h_i[k]\right]$.

The difference in expected margin losses between the source and target domains is given by:

$$\mathcal{L}_{\gamma/2}^\mathsf{T}(h) - \mathcal{L}_\gamma^\mathsf{S}(h) = \mathbb{E}_{y^\mathsf{T}}\left[\frac{1}{N_\mathsf{T}} \sum_{j \in \mathcal{V}^\mathsf{T}} \mathcal{L}_{\gamma/2}(h_j, y_j)\right] - \mathbb{E}_{y^\mathsf{S}}\left[\frac{1}{N_\mathsf{S}} \sum_{i \in \mathcal{V}^\mathsf{S}} \mathcal{L}_\gamma(h_i, y_i)\right]. \tag{19}$$

Using Definition 1, Eq. (19) can be rewritten and bounded as follows:

$$\leq \frac{1}{\max(N_\mathsf{S}, N_\mathsf{T})} \sum_{i \in \mathcal{V}^\mathsf{S}} \frac{1}{N_\mathsf{T}} \left(\sum_{j \in \mathcal{V}^\mathsf{T}} \mathbb{E}_{y_j} \mathcal{L}_{\gamma/2}(h_j, y_j) - \mathbb{E}_{y_i} \mathcal{L}_\gamma(h_i, y_i)\right) \tag{20}$$

$$= \frac{1}{\max(N_\mathsf{S}, N_\mathsf{T})} \sum_{i \in \mathcal{V}^\mathsf{S}} \frac{1}{N_\mathsf{T}} \sum_{j \in \mathcal{V}^\mathsf{T}} \sum_k \left(\eta_k(j)\mathcal{L}_{\gamma/2}(h_j, k) - \mathbb{P}(y_i = k)\mathcal{L}_\gamma(h_i, k)\right) \tag{21}$$

$$= \frac{1}{\max(N_\mathsf{S}, N_\mathsf{T})} \sum_{i \in \mathcal{V}^\mathsf{S}} \frac{1}{N_\mathsf{T}} \sum_{j \in \mathcal{V}^\mathsf{T}} \sum_k \left(\eta_k(j)\mathcal{L}_{\gamma/2}(h_j, k) - \eta_k(i)\mathcal{L}_\gamma(h_i, k)\right) \tag{22}$$

$$= \frac{1}{\max(N_\mathsf{S}, N_\mathsf{T})} \sum_{i \in \mathcal{V}^\mathsf{S}} \frac{1}{N_\mathsf{T}} \sum_{j \in \mathcal{V}^\mathsf{T}} \sum_k \left(\eta_k(j)\left(\mathcal{L}_{\gamma/2}(h_j, k) - \mathcal{L}_\gamma(h_i, k)\right) + (\eta_k(j) - \eta_k(i))\mathcal{L}_\gamma(h_i, k)\right) \tag{23}$$

$$\leq \frac{1}{\max(N_\mathsf{S}, N_\mathsf{T})} \sum_{i \in \mathcal{V}^\mathsf{S}} \frac{1}{N_\mathsf{T}} \sum_{j \in \mathcal{V}^\mathsf{T}} \sum_k \left(\mathcal{L}_{\gamma/2}(h_j, k) - \mathcal{L}_\gamma(h_i, k) + \|\eta_k(j) - \eta_k(i)\|_2^2\right). \tag{24}$$

Now, applying Theorem 4 to bound the difference in conditional probabilities, we obtain:

$$\sum_k \|\eta_k(j) - \eta_k(i)\|_2^2 \tag{25}$$

$$\leq \mathcal{O}\left(\mathbb{E}\left[\|Z_u - Z_v\|_2^2\right] + \sum_{c=0}^k \left(\text{Var}_\mathsf{S}(Z_u \mid y_u = c) + \text{Var}_\mathsf{T}(Z_v \mid y_v = c) + \|\boldsymbol{\mu}_c^{(\mathsf{S})} - \boldsymbol{\mu}_c^{(\mathsf{T})}\|_2^2\right)\right), \tag{26}$$

where $\boldsymbol{\mu}_c^{(\mathsf{S})}$ and $\boldsymbol{\mu}_c^{(\mathsf{T})}$ are the class-conditional means for the source and target domains, respectively. This bound is second-order in nature, as it controls the discrepancy through pairwise distances and class-conditional variances or means. In principle, one could generalize this to a $K$-th order metric by incorporating higher-order moments (e.g., skewness (Seglen, 1992), kurtosis (Balanda & MacGillivray, 1988)). However, such higher-order terms quickly become cumbersome and may lead to instability in high-dimensional settings. To handle this in a more unified and tractable manner, we utilize characteristic function, which encode moments of all orders and can uniquely determine a distribution. The characteristic function for a random vector $\mathbf{X}$ is defined as:

$$\Psi_{\mathbf{X}}(\mathbf{t}) = \mathbb{E}\left[e^{i\mathbf{t}^\top \mathbf{X}}\right], \tag{27}$$

which can be expanded into a power series:

$$\Psi_{\mathbf{X}}(\mathbf{t}) = \sum_{k=0}^\infty \frac{i^k}{k!}(\mathbf{t}^\top)^k \mathbb{E}[\mathbf{X}^{\otimes k}]. \tag{28}$$

This expansion makes clear that the characteristic function encapsulates moments of all orders, which naturally subsumes the information contained in second-order bounds while remaining computationally efficient for distributional alignment. Motivated by this representation, we lift the second-order bound to a distributional discrepancy on the complex plane via CF:

$$\mathsf{D}_{\gamma/2}^{\mathsf{S},\mathsf{T}}(\mathbb{P}; \xi) \leq \mathcal{O}\left(\sum_{m=1}^M \omega_m \left|\Psi^\mathsf{S}(\mathbf{t}_m) - \Psi^\mathsf{T}(\mathbf{t}_m)\right|^2\right). \tag{29}$$

---

**Algorithm 1:** minimax Optimization for Graph Domain Adaptation

---

**Input:** Source graph $\mathcal{G}^{\mathsf{S}} = (\mathcal{X}^{\mathsf{S}}, \mathcal{A}^{\mathsf{S}}, \mathcal{Y}^{\mathsf{S}})$, target graph $\mathcal{G}^{\mathsf{T}} = (\mathcal{X}^{\mathsf{T}}, \mathcal{A}^{\mathsf{T}})$, trade-off parameter $\lambda$,
        learning rates $\eta_\delta, \eta_\phi$.
**Output:** Trained GNN parameters $\boldsymbol{\delta}$, frequency selector parameters $\boldsymbol{\phi}$.
Initialize $\boldsymbol{\delta}$, $\boldsymbol{\phi}$ randomly;
**for** *each epoch* $= 1, \ldots, T_{train}$ **do**
   | Sample mini-batch from $\mathcal{G}^{\mathsf{S}}$ and $\mathcal{G}^{\mathsf{T}}$;
   | `/* Step A: Update GNN parameters `$\boldsymbol{\delta}$` (fix `$\boldsymbol{\phi}$`)`             `*/`
   | Compute source loss: $\mathcal{L}_{\text{source}}(\boldsymbol{\delta}) = \text{CE}\big(\text{GNN}_{\boldsymbol{\delta}}(\mathcal{X}^{\mathsf{S}}, \mathcal{A}^{\mathsf{S}}), \mathcal{Y}^{\mathsf{S}}\big)$;
   | Compute alignment loss: $\mathcal{L}_{\text{align}}(\boldsymbol{\delta}, \boldsymbol{\phi}) = \int \sqrt{\ell_\kappa(\mathbf{t}; \boldsymbol{\delta})}\, p_{\mathcal{T}}(\mathbf{t}; \boldsymbol{\phi})\, d\mathbf{t}$;
   | Total loss: $\mathcal{L} = \mathcal{L}_{\text{source}} + \lambda \mathcal{L}_{\text{align}}$;
   | Update $\boldsymbol{\delta} \leftarrow \boldsymbol{\delta} - \eta_\delta \nabla_{\boldsymbol{\delta}} \mathcal{L}$;
   | `/* Step B: Update frequency sampler `$\boldsymbol{\phi}$` (fix `$\boldsymbol{\delta}$`)`        `*/`
   | Compute alignment loss with fixed $\boldsymbol{\delta}$;
   | Update $\boldsymbol{\phi} \leftarrow \boldsymbol{\phi} + \eta_\phi \nabla_{\boldsymbol{\phi}} \mathcal{L}_{\text{align}}$
**return** $\boldsymbol{\delta}, \boldsymbol{\phi}$.

---

This bound suggests that the domain discrepancy can be controlled by a weighted sum of squared differences between the characteristic functions of the source and target domains, evaluated at multiple frequency points. The weights $\omega_m$ allow for adaptive weighting of different frequency components, reflecting the intuition that some frequencies may be more important for generalization than others.

Our proposed adaptive frequency selection mechanism directly leverages this insight. The weights $\omega_m$ are learned to emphasize frequencies where significant cross-domain discrepancies exist. This allows the model to focus on discriminative features while suppressing irrelevant or noisy components, leading to more robust and effective domain adaptation.      □

**Remark.** To clarify the relationship between the learned frequency distribution $p_{\mathcal{T}}(\mathbf{t}; \boldsymbol{\phi})$ and the discrepancy weights $\omega_m$, we observe that the adaptive frequency sampler effectively prioritizes the most relevant frequencies based on the transfer scenario. Specifically, the optimization of the alignment loss with respect to the $p_{\mathcal{T}}(\mathbf{t}; \boldsymbol{\phi})$ is equivalent to or upper-bounded by maximizing a weighted sum of individual frequency losses, where the weights $\omega_m$ correspond to the importance of each frequency in driving the overall distributional alignment. This relationship can be formalized through importance sampling, where $p_{\mathcal{T}}(\mathbf{t}; \boldsymbol{\phi})$ adjusts the sampling probability of each frequency, naturally aligning the most relevant features without requiring explicit weight assignments.

## C    TRAINING DETAILS

### C.1    COMPLEXITY ANALYSIS

In this section, we analyze the time complexity of the proposed ADAlign framework. For one training iteration, the computational cost mainly comes from GNN message passing and spectral alignment. Suppose the source and target graphs together have $|\mathcal{V}|$ nodes and $|\mathcal{E}|$ edges, the embedding dimension is $d$, the number of layers is $L$, and $M$ frequency samples are used in alignment. The GNN forward and backward passes take $\mathcal{O}\big(L(|\mathcal{E}|\,d + |\mathcal{V}|\,d^2)\big)$, reflecting sparse matrix multiplications and linear transformations at each layer. The spectral alignment requires computing $M$ frequency responses for all node embeddings, each involving a $d$-dimensional inner product, resulting in $\mathcal{O}(M|\mathcal{V}|d)$. Thus, the overall per-iteration complexity is $\mathcal{O}\big(L(|\mathcal{E}|\,d + |\mathcal{V}|\,d^2)\big) + \mathcal{O}(M|\mathcal{V}|d)$, which is linear in the graph size $|\mathcal{V}|, |\mathcal{E}|$ and at most quadratic in the embedding dimension $d$.

### C.2    ALGORITHM OF ADALIGN

The complete training procedure of our framework is summarized in Algorithm 1.

| Types | # Datasets | # Nodes | # Edges | # Feat Dims | # Labels |
|---|---|---|---|---|---|
| ArnetMiner | ACMv9 (A) | 9,360 | 31,112 | 6,775 | 5 |
| | Citationv1 (C) | 8,935 | 30,196 | | |
| | DBLPv7 (D) | 5,484 | 16,234 | | |
| Airport | USA (U) | 1,190 | 27,198 | 241 | 4 |
| | Brazil (B) | 131 | 2,148 | | |
| | Europe (E) | 399 | 11,990 | | |
| Twitch | England (EN) | 7,126 | 35,324 | 3,170 | 2 |
| | Germany (DE) | 9,498 | 153,138 | | |
| Blog | Blog1 (B1) | 2,300 | 66,942 | 8,189 | 6 |
| | Blog2 (B2) | 2,896 | 107,672 | | |

Table 3: Dataset statistics.

## D EXPERIMENTAL SETUP DETAILS

### D.1 DATASET DESCRIPTION

We conduct node classification experiments across four distinct migration settings, as outlined in Table 3. In each setting, our model is trained on one graph and its performance is evaluated on the remaining graphs. The datasets used in these experiments are as follows:

- ACMv9 (A), Citationv1 (C), DBLPv7 (D): These datasets are citation networks collected from distinct academic sources. In these networks, each node represents a research paper, and the edges denote citation relationships between papers. The goal of the task is to classify each research paper into its corresponding category. The datasets are sourced from different time periods and repositories: ACM (before 2008), DBLP (from 2004 to 2008), and Microsoft Academic Graph (after 2010). We perform experiments under six different transfer scenarios, namely: A → C, A → D, C → A, C → D, D → A, and D → C.

- USA (U), Brazil (B), Europe (E): These datasets are derived from transportation statistics, specifically focusing on aviation data. In these datasets, each node corresponds to an airport, and the edges represent direct flight connections between two airports. The objective of the task is to predict the category of each airport based on its characteristics and network relationships. We explore the following six transfer scenarios: U → B, U → E, B → U, B → E, E → U, and E → B.

- Blog1 (B1), Blog2 (B2): These datasets come from the BlogCatalog dataset, where each node represents a blogger, and the edges represent the friendship relationships between bloggers. The node attributes consist of keywords extracted from the bloggers' self-descriptions, and the task is to predict the group or category to which each blogger belongs. We set up two specific transfer scenarios: B1 → B2 and B2 → B1.

- Germany (DE), England (EN): These datasets are sourced from the Twitch Social Networks, where nodes represent streamers, and edges denote the follower relationships between streamers. Each node is associated with attributes such as categories or tags that describe the streamer's profile. The task is to predict the category of each streamer based on their attributes and relationships within the network. We investigate two migration scenarios: DE → EN and EN → DE.

### D.2 BASELINES

We compare our method with the following baseline methods, which can be divided into two categories: (1) Graph Neural Networks (GNNs) on the source graph only: GAT (Veličković et al., 2018), GIN (Xu et al., 2019), and GCN (Kipf & Welling, 2017) are trained end-to-end on the source graph, allowing direct application to the target graph; (2) Graph Domain Adaptation Methods: DANE (Zhang et al., 2019), UDAGCN (Wu et al., 2020), AdaGCN (Dai et al., 2022), StruRW (Liu et al., 2023), SA-GDA (Pang et al., 2023), GRADE (Wu et al., 2023), PairAlign (Liu et al., 2024b), GraphAlign (Huang et al., 2024), A2GNN (Liu et al., 2024a), TDSS (Chen et al., 2025a), DGSDA (Yang et al., 2025), GAA (Fang et al., 2025a) and HGDA (Fang et al., 2025b) are specifically designed to address the GDA problem.

Below, we provide descriptions of the baselines used in the experiments.

- **ERM (GAT) , ERM (GIN) , ERM (GCN)** . These methods train using GAT (Graph Attention Networks), GIN (Graph Isomorphism Networks), and GCN (Graph Convolution Networks) under the standard Empirical Risk Minimization (ERM) framework.

- **DANE** employs shared-weight GCNs to generate node representations. To address distribution shifts, it leverages a least-squares generative adversarial network, which promotes alignment between the source and target distributions of node representations.

- **UDAGCN** is a model-centric method that combines adversarial learning with GNNs for domain adaptation. It aims to reduce the discrepancy between source and target graphs by using an adversarial objective, promoting the learning of domain-invariant features across different domains.

- **AdaGCN** is a model-centric method designed for GDA. It incorporates adversarial domain adaptation techniques along with graph convolution to align node representations across domains, using the empirical Wasserstein distance as a regularization term to minimize distribution shifts between the source and target domains.

- **STRURW** modifies graph data by adjusting edge weights based on domain adaptation strategies. This adjustment is informed by the data-centric GDA techniques, ensuring that the model effectively adapts to distribution shifts while preserving the structural properties of the graph.

- **SA-GDA** observes that nodes from the same class across different domains share similar spectral characteristics, while different classes remain spectrally separable. Based on this, it aligns source-target feature spaces in the spectral domain at the category level, avoiding confusion caused by global feature alignment.

- **GRADE** introduces a graph subtree discrepancy metric to measure distribution shifts. By connecting graph neural networks with the Weisfeiler-Lehman subtree kernel, it evaluates how node embeddings differ across domains, enabling more effective transfer under distributional variations.

- **PairAlign** is a method that recalibrates the influence between neighboring nodes using edge weights to address conditional structure shifts. Additionally, it adjusts the classification loss with label weights to account for label shifts, ensuring that the model can handle both structural and label distribution shifts effectively.

- **GraphAlign** generates a small, transferable graph from the source graph, which is then used to train a GNN under Empirical Risk Minimization (ERM). This approach focuses on leveraging smaller graphs to improve domain adaptation by aligning the distribution of node representations across domains while preserving graph structure.

- **A2GNN** proposes a simple yet effective GNN architecture that increases the number of propagation layers in the target graph. This increase in depth allows the model to capture complex dependencies within target domain, improving its adaptation performance in domain-shifted tasks.

- **TDSS** applies structural smoothing directly to the target graph to mitigate local variations. This method helps enhance model robustness by preserving consistency in node representations, which improves the transfer of knowledge across domains while reducing the impact of structural discrepancies between the source and target graphs.

- **DGSDA** addresses the challenge of domain adaptation for graph-structured data by disentangling the alignment of node attributes and graph topology. This framework uses Bernstein polynomial approximation to align graph spectral filters efficiently, avoiding expensive eigenvalue decomposition. By separating attribute and topology shifts, DGSDA enables the use of flexible GNNs, improving model performance and adaptability across domains.

- **GAA** focuses on addressing both graph topology and node attribute shifts in Graph Domain Adaptation (GDA). Unlike traditional methods that primarily address topology discrepancies, GAA emphasizes the critical role of node attributes in aligning domain-specific graph features. The method introduces a cross-channel module that uses attention-based learning for attribute alignment, allowing the model to focus on significant attributes.

- **HGDA** highlights the impact of homophily distribution discrepancies between source and target graphs in GDA tasks. By investigating the homophilic and heterophilic patterns in graph nodes, HGDA proposes a novel method that uses mixed filters to capture and mitigate homophily shifts. The method separates homophilic, heterophilic, and attribute signals in graphs, enhancing domain alignment across different graph types.

### D.3 Implementation Details

We utilize Micro-F1 and Macro-F1 as evaluation metrics for our experiments. For each experimental result, we conducted five runs and calculated the average performance across these runs. Each run was executed for a total of 150 epochs. we train our model by utilizing the Adam (Kingma & Ba., 2015) optimizer with learning rate ranging from 0.001 to 0.01. The hyperparameter search ranges are as follows: the dropout rate is in the range $[0.1, 0.5]$, the propagation steps take values from the set $\{0, 1, 10, 15\}$, the frequency number is chosen from $\{32, 64, 128, 256, 512, 1024, 2048, 4096\}$, the amplitude weight coefficient $\kappa$ is in the range $[0, 1]$.The experiments were conducted on a Linux server , running Ubuntu 18.04.6 LTS. For GPU resources, we utilized a single NVIDIA GeForce RTX 3090 graphics card with 24GB of memory. The Python libraries employed for implementing our experiments include Python 3.8, PyTorch 2.4.0, PyTorch Geometric 2.6.1, PyTorch Sparse 0.6.18, and PyTorch Scatter 2.1.2. The complete parameter space is provided in Table 7.

## E  Experimental Results

| Methods | $A \to C$ | $A \to D$ | $C \to A$ | $C \to D$ | $D \to A$ | $D \to C$ | $B1 \to B2$ | $B2 \to B1$ |
|---|---|---|---|---|---|---|---|---|
| GAT (ICLR'18) | $59.60_{\pm 2.82}$ | $55.09_{\pm 1.16}$ | $58.06_{\pm 1.88}$ | $63.82_{\pm 1.85}$ | $52.99_{\pm 2.36}$ | $59.96_{\pm 2.86}$ | $11.18_{\pm 3.84}$ | $9.43_{\pm 3.04}$ |
| GIN (ICLR'19) | $62.34_{\pm 1.08}$ | $53.08_{\pm 2.06}$ | $61.34_{\pm 0.62}$ | $64.83_{\pm 0.44}$ | $51.36_{\pm 2.36}$ | $63.09_{\pm 1.92}$ | $11.44_{\pm 2.68}$ | $9.57_{\pm 0.28}$ |
| GCN (ICLR'17) | $66.49_{\pm 2.21}$ | $59.53_{\pm 0.44}$ | $65.06_{\pm 1.38}$ | $65.80_{\pm 1.82}$ | $58.95_{\pm 0.57}$ | $64.66_{\pm 1.01}$ | $13.80_{\pm 1.15}$ | $13.76_{\pm 1.33}$ |
| DANE (IJCAI'19) | $66.89_{\pm 2.90}$ | $59.09_{\pm 2.60}$ | $64.04_{\pm 2.01}$ | $60.34_{\pm 3.01}$ | $57.98_{\pm 1.47}$ | $62.37_{\pm 2.47}$ | $25.20_{\pm 2.13}$ | $28.37_{\pm 0.89}$ |
| UDAGCN (WWW'20) | $78.74_{\pm 0.55}$ | $72.59_{\pm 1.05}$ | $74.37_{\pm 0.35}$ | $\underline{75.56}_{\pm 0.42}$ | $70.10_{\pm 0.60}$ | $76.09_{\pm 0.78}$ | $28.83_{\pm 1.15}$ | $28.95_{\pm 0.36}$ |
| AdaGCN (TKDE'22) | $64.15_{\pm 1.88}$ | $61.97_{\pm 1.25}$ | $63.22_{\pm 1.25}$ | $66.43_{\pm 0.43}$ | $58.99_{\pm 1.42}$ | $58.27_{\pm 1.49}$ | $20.69_{\pm 1.40}$ | $18.73_{\pm 2.91}$ |
| StruRW (ICML'23) | $54.49_{\pm 1.41}$ | $52.54_{\pm 0.57}$ | $51.77_{\pm 2.01}$ | $56.74_{\pm 2.01}$ | $44.68_{\pm 1.25}$ | $50.47_{\pm 1.25}$ | $39.62_{\pm 0.77}$ | $41.36_{\pm 0.47}$ |
| SA-GDA (MM'23) | $72.10_{\pm 2.29}$ | $65.13_{\pm 3.98}$ | $71.40_{\pm 1.99}$ | $\underline{70.97}_{\pm 2.78}$ | $56.83_{\pm 1.90}$ | $69.08_{\pm 2.86}$ | $40.26_{\pm 2.31}$ | $\underline{44.59}_{\pm 1.77}$ |
| GRADE (AAAI'23) | $71.66_{\pm 0.50}$ | $63.05_{\pm 0.41}$ | $68.43_{\pm 0.14}$ | $69.76_{\pm 0.92}$ | $56.45_{\pm 0.60}$ | $66.54_{\pm 0.75}$ | $44.46_{\pm 4.01}$ | $42.00_{\pm 1.52}$ |
| PairAlign (ICML'24) | $55.11_{\pm 1.02}$ | $53.80_{\pm 0.85}$ | $51.57_{\pm 1.36}$ | $59.02_{\pm 0.82}$ | $46.47_{\pm 1.13}$ | $53.66_{\pm 0.47}$ | $39.57_{\pm 0.85}$ | $41.98_{\pm 0.85}$ |
| GraphAlign (KDD'24) | $71.09_{\pm 1.21}$ | $65.51_{\pm 1.21}$ | $61.37_{\pm 0.28}$ | $71.18_{\pm 1.32}$ | $62.33_{\pm 0.45}$ | $65.56_{\pm 0.21}$ | $43.14_{\pm 0.55}$ | $42.29_{\pm 3.28}$ |
| A2GNN (AAAI'24) | $78.06_{\pm 0.73}$ | $71.31_{\pm 0.34}$ | $\underline{76.41}_{\pm 0.54}$ | $73.28_{\pm 0.32}$ | $\underline{74.48}_{\pm 0.68}$ | $76.01_{\pm 0.79}$ | $45.42_{\pm 4.75}$ | $43.14_{\pm 1.77}$ |
| TDSS (AAAI'25) | $\underline{79.10}_{\pm 1.64}$ | $71.59_{\pm 3.08}$ | $76.09_{\pm 1.06}$ | $74.96_{\pm 0.75}$ | $73.23_{\pm 3.09}$ | $\underline{77.70}_{\pm 2.68}$ | $\underline{48.31}_{\pm 2.00}$ | $44.31_{\pm 1.89}$ |
| DGSDA (ICML'25) | $77.03_{\pm 0.52}$ | $\underline{72.63}_{\pm 0.25}$ | $71.61_{\pm 0.80}$ | $75.52_{\pm 0.34}$ | $71.82_{\pm 0.45}$ | $76.13_{\pm 0.23}$ | $48.01_{\pm 1.71}$ | $42.06_{\pm 3.36}$ |
| GAA (ICLR'25) | $75.85_{\pm 0.65}$ | $68.07_{\pm 1.09}$ | $73.57_{\pm 0.55}$ | $71.40_{\pm 0.16}$ | $62.66_{\pm 0.93}$ | $72.74_{\pm 0.76}$ | $45.66_{\pm 2.12}$ | $40.59_{\pm 2.04}$ |
| HGDA (ICML'25) | $74.28_{\pm 0.71}$ | $69.03_{\pm 4.12}$ | $66.98_{\pm 0.97}$ | $70.59_{\pm 0.46}$ | $66.47_{\pm 0.97}$ | $72.85_{\pm 0.82}$ | $45.32_{\pm 2.04}$ | $43.35_{\pm 1.21}$ |
| **ADAlign (Ours)** | $\mathbf{80.65}_{\pm 0.36}$ | $\mathbf{76.00}_{\pm 1.07}$ | $\mathbf{77.62}_{\pm 0.25}$ | $\mathbf{77.48}_{\pm 0.35}$ | $\mathbf{75.83}_{\pm 0.93}$ | $\mathbf{79.95}_{\pm 0.45}$ | $\mathbf{54.89}_{\pm 0.59}$ | $\mathbf{51.85}_{\pm 0.26}$ |

| Methods | $U \to B$ | $U \to E$ | $B \to U$ | $B \to E$ | $E \to U$ | $E \to B$ | $GE \to EN$ | $EN \to GE$ |
|---|---|---|---|---|---|---|---|---|
| GAT (ICLR'18) | $48.01_{\pm 4.03}$ | $35.57_{\pm 3.78}$ | $41.25_{\pm 1.57}$ | $46.18_{\pm 1.23}$ | $42.13_{\pm 1.72}$ | $49.02_{\pm 4.50}$ | $57.22_{\pm 0.20}$ | $57.69_{\pm 1.18}$ |
| GIN (ICLR'19) | $23.29_{\pm 4.55}$ | $22.50_{\pm 4.12}$ | $28.06_{\pm 1.93}$ | $26.03_{\pm 4.30}$ | $28.03_{\pm 2.24}$ | $26.21_{\pm 4.23}$ | $51.96_{\pm 2.70}$ | $50.32_{\pm 4.60}$ |
| GCN (ICLR'17) | $55.12_{\pm 5.22}$ | $42.63_{\pm 1.04}$ | $31.21_{\pm 2.51}$ | $46.52_{\pm 1.77}$ | $37.39_{\pm 4.21}$ | $57.27_{\pm 3.13}$ | $56.71_{\pm 0.41}$ | $58.43_{\pm 0.47}$ |
| DANE (IJCAI'19) | $41.23_{\pm 1.63}$ | $34.24_{\pm 4.76}$ | $35.73_{\pm 5.31}$ | $33.57_{\pm 3.90}$ | $28.26_{\pm 7.44}$ | $39.94_{\pm 0.80}$ | $48.80_{\pm 4.72}$ | $49.01_{\pm 2.59}$ |
| UDAGCN (WWW'20) | $56.82_{\pm 2.59}$ | $41.85_{\pm 0.86}$ | $34.65_{\pm 1.26}$ | $47.60_{\pm 1.75}$ | $41.83_{\pm 1.07}$ | $61.97_{\pm 4.15}$ | $58.31_{\pm 0.90}$ | $59.45_{\pm 1.33}$ |
| AdaGCN (TKDE'22) | $66.15_{\pm 1.38}$ | $\underline{51.45}_{\pm 0.93}$ | $43.89_{\pm 1.07}$ | $\underline{54.92}_{\pm 0.91}$ | $44.37_{\pm 0.64}$ | $\underline{73.34}_{\pm 2.46}$ | $56.68_{\pm 0.19}$ | $59.86_{\pm 0.43}$ |
| StruRW (ICML'23) | $62.25_{\pm 2.61}$ | $37.26_{\pm 2.15}$ | $40.02_{\pm 1.56}$ | $36.84_{\pm 1.54}$ | $38.10_{\pm 0.78}$ | $52.15_{\pm 0.57}$ | $52.76_{\pm 2.63}$ | $50.51_{\pm 2.84}$ |
| SA-GDA (MM'23) | $\underline{66.51}_{\pm 1.87}$ | $50.39_{\pm 0.69}$ | $47.94_{\pm 2.86}$ | $53.92_{\pm 1.24}$ | $45.34_{\pm 0.79}$ | $64.63_{\pm 1.16}$ | $54.20_{\pm 2.41}$ | $57.24_{\pm 0.82}$ |
| GRADE (AAAI'23) | $56.16_{\pm 0.96}$ | $46.86_{\pm 0.22}$ | $38.68_{\pm 0.41}$ | $53.93_{\pm 0.33}$ | $42.47_{\pm 0.65}$ | $60.13_{\pm 0.36}$ | $58.36_{\pm 0.06}$ | $59.62_{\pm 0.13}$ |
| PairAlign (ICML'24) | $65.35_{\pm 2.65}$ | $35.61_{\pm 0.45}$ | $40.89_{\pm 2.33}$ | $37.57_{\pm 3.00}$ | $37.71_{\pm 0.48}$ | $53.26_{\pm 1.27}$ | $55.04_{\pm 0.14}$ | $60.68_{\pm 0.28}$ |
| GraphAlign (KDD'24) | $61.33_{\pm 1.21}$ | $50.09_{\pm 2.33}$ | $47.11_{\pm 1.17}$ | $53.19_{\pm 0.78}$ | $\underline{49.31}_{\pm 0.37}$ | $69.97_{\pm 1.22}$ | $48.14_{\pm 0.57}$ | $59.33_{\pm 0.44}$ |
| A2GNN (AAAI'24) | $61.08_{\pm 2.57}$ | $43.51_{\pm 3.94}$ | $38.60_{\pm 7.73}$ | $42.80_{\pm 3.55}$ | $48.39_{\pm 1.07}$ | $46.76_{\pm 2.47}$ | $54.25_{\pm 0.88}$ | $51.78_{\pm 3.16}$ |
| TDSS (AAAI'25) | $63.36_{\pm 1.33}$ | $48.17_{\pm 1.50}$ | $36.46_{\pm 2.63}$ | $40.38_{\pm 1.15}$ | $36.55_{\pm 2.14}$ | $45.27_{\pm 1.12}$ | $55.01_{\pm 0.87}$ | $53.19_{\pm 4.24}$ |
| DGSDA (ICML'25) | $60.28_{\pm 1.55}$ | $48.20_{\pm 1.46}$ | $\underline{53.76}_{\pm 0.78}$ | $42.58_{\pm 1.73}$ | $47.59_{\pm 0.76}$ | $59.30_{\pm 0.63}$ | $\underline{59.76}_{\pm 0.12}$ | $\underline{61.58}_{\pm 0.24}$ |
| GAA (ICLR'25) | $59.28_{\pm 2.67}$ | $50.16_{\pm 1.01}$ | $50.18_{\pm 1.03}$ | $42.76_{\pm 0.64}$ | $47.17_{\pm 1.30}$ | $55.34_{\pm 2.73}$ | $56.88_{\pm 0.20}$ | $59.45_{\pm 0.92}$ |
| HGDA (ICML'25) | $63.27_{\pm 1.94}$ | $44.48_{\pm 1.96}$ | $50.25_{\pm 2.51}$ | $43.17_{\pm 0.45}$ | $43.44_{\pm 3.93}$ | $56.74_{\pm 2.06}$ | $56.61_{\pm 0.44}$ | $54.84_{\pm 2.84}$ |
| **ADAlign (Ours)** | $\mathbf{71.96}_{\pm 1.40}$ | $\mathbf{52.74}_{\pm 0.52}$ | $\mathbf{54.47}_{\pm 0.23}$ | $\mathbf{59.10}_{\pm 1.13}$ | $\mathbf{51.37}_{\pm 0.75}$ | $\mathbf{75.74}_{\pm 1.09}$ | $\mathbf{60.77}_{\pm 0.54}$ | $\mathbf{62.11}_{\pm 0.29}$ |

Table 4: Node classification performance ($\% \pm \sigma$, Macro-F1 score) under 16 transfer scenarios. The highest scores are highlighted in **bold**, while the second-highest scores are underlined.

### E.1 Main Experimental Results

The macro-F1 scores are reported in Table 4.

### E.2 SIGNIFICANCE TEST (T-TEST) OF ADALIGN

We conduct paired two-sided t-tests comparing ADAlign with all baselines across A→C and A→D. In Figure 8 and Figure 9 , all bars lie well above the $-\log_2(0.05)$ threshold, indicating statistically significant improvements ($p < 0.05$) in every setting.

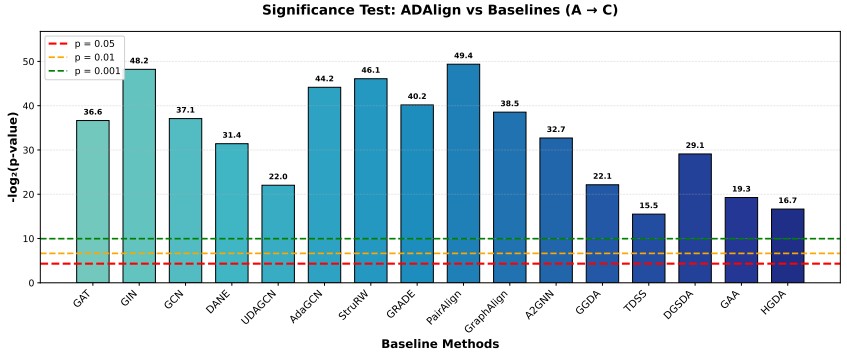

Figure 8: t-test on the A→C task with significance level 0.05.

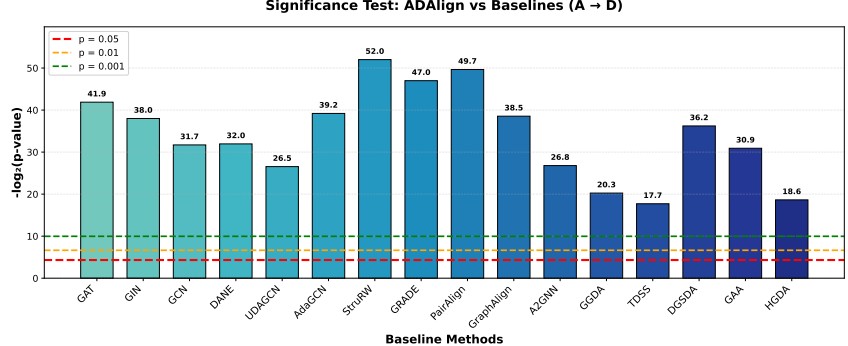

Figure 9: t-test on the A→D task with significance level 0.05.

### E.3 EFFECT OF GNN BACKBONES

To examine the influence of different GNN backbones, we instantiated ADAlign on two widely used architectures (GCN and GAT) and evaluated it against strong baselines on two transfer scenarios. As shown in Table 5 and Table 6, ADAlign consistently outperforms the corresponding source-only models and all competing methods under both backbones, often by a clear margin. These findings demonstrate that ADAlign is not tied to a particular encoder and can be reliably integrated with diverse GNN architectures.

| Method | A→C (Mi-F1) | A→C (Ma-F1) | A→D (Mi-F1) | A→D (Ma-F1) |
|---|---|---|---|---|
| GCN (ICLR'17) | $70.82 \pm 1.26$ | $66.49 \pm 2.21$ | $65.05 \pm 2.15$ | $59.53 \pm 0.44$ |
| GAA (ICLR'25) | $73.50 \pm 0.85$ | $70.20 \pm 1.30$ | $67.30 \pm 1.50$ | $62.80 \pm 0.90$ |
| HGDA (ICML'25) | $75.40 \pm 0.60$ | $72.80 \pm 0.90$ | $68.65 \pm 1.10$ | $64.50 \pm 1.15$ |
| **ADAlign (Ours)** | **$76.99 \pm 0.37$** | **$75.11 \pm 0.45$** | **$70.11 \pm 0.80$** | **$66.33 \pm 1.33$** |

Table 5: Results on the GCN backbone.

| Method | A→C (Mi-F1) | A→C (Ma-F1) | A→D (Mi-F1) | A→D (Ma-F1) |
|---|---|---|---|---|
| GAT (ICLR'18) | $62.77 \pm 2.24$ | $59.60 \pm 2.82$ | $60.29 \pm 1.33$ | $55.09 \pm 1.16$ |
| GAA (ICLR'25) | $67.63 \pm 1.42$ | $64.53 \pm 1.74$ | $63.33 \pm 1.03$ | $59.26 \pm 2.06$ |
| HGDA (ICML'25) | $65.20 \pm 1.80$ | $62.10 \pm 2.20$ | $61.80 \pm 1.20$ | $57.30 \pm 1.60$ |
| **ADAlign (Ours)** | $\mathbf{70.45 \pm 0.95}$ | $\mathbf{69.80 \pm 1.10}$ | $\mathbf{67.45 \pm 0.85}$ | $\mathbf{65.90 \pm 1.25}$ |

Table 6: Results on the GAT backbone.

### E.4 LOSS COMPARISON

To evaluate the stability of ADAlign, we track the training loss over epochs for A→C and A→D. As shown in Figure 10, the loss decreases smoothly and rapidly during the early epochs, eventually plateauing at a low level with minimal oscillations. This indicates a well-behaved and stable optimization process. Notably, both transfers exhibit nearly identical convergence trajectories, further demonstrating that ADAlign maintains stability across different domain shifts.

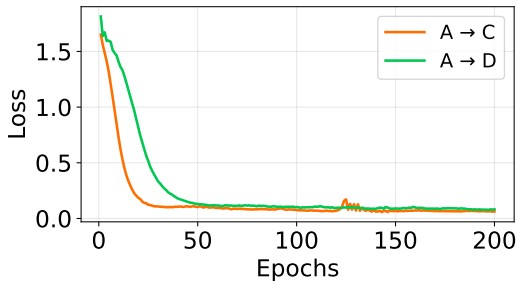

Figure 10: Training loss curves of ADAlign on two tasks.

### E.5 ABLATION STUDY

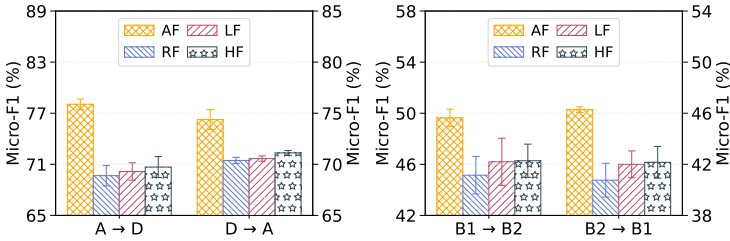

Figure 11: Classification Mi-F1 comparisons between ADAlign variants on four cross-domain tasks.

### E.6 PARAMETER SENSITIVITY ANALYSIS

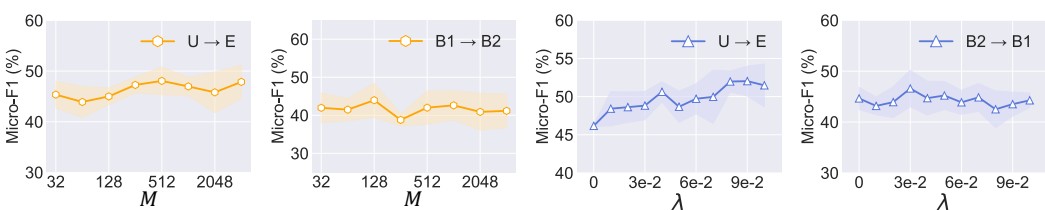

Figure 12: The performances of our ADAlign w.r.t varying parameters on different transfer tasks

### E.7 Visualization of the learned frequency distributions

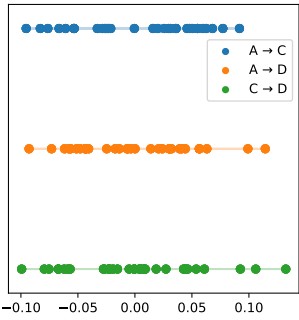

Figure 13: Learned frequency sampling distributions for three transfer scenarios.

In this section, we visualize the learned frequency distributions across several transfer scenarios. As shown in Figure 13, ADAlign learns distinct dominant frequency regions for different datasets, confirming that the adaptive sampler does not rely on a fixed or heuristic grid. Instead, it automatically emphasizes spectral components that reveal the most salient discrepancies for each source–target pair. For example, the A→C scenario concentrates more probability mass on higher-frequency components, whereas A→D and C→D favor lower or mid-range frequencies. These patterns support the claim that ADAlign tailors its spectral emphasis to the underlying structural and attribute-level shifts of each task, aligning with the motivation of NSD and its minimax training strategy.

| Dataset | Transfer Scenarios | Hyperparameter | Search Space |
|---|---|---|---|
|  | A → C | learning rate | [0.001, 0.005, 0.01] |
|  |  | weight decay | [0.001, 0.005, 0.01] |
|  |  | frequency number | [ 128, 1024, 2048, 4096] |
|  |  | pnums | [0, 1, 10, 15] |
|  |  | lambda | [0, 0.2, 0.4, 0.6, 0.8, 1] |
|  |  | dropout ratio | [0.1, 0.25, 0.5] |
|  |  | feature dimension | 128 |
|  |  | amplitude weight | $0 \sim 1$ |
|  |  | epochs | 150 |
|  | A → D | learning rate | [0.001, 0.005, 0.01] |
|  |  | weight decay | [0.001, 0.005, 0.01] |
|  |  | frequency number | [ 128, 1024, 2048, 4096] |
|  |  | pnums | [0, 1, 10, 15] |
|  |  | lambda | [0, 0.2, 0.4, 0.6, 0.8, 1] |
|  |  | dropout ratio | [0.1, 0.25, 0.5] |
|  |  | feature dimension | 128 |
|  |  | amplitude weight | $0 \sim 1$ |
|  |  | epochs | 150 |
|  | C → A | learning rate | [0.001, 0.005, 0.01] |
|  |  | weight decay | [0.001, 0.005, 0.01] |
|  |  | frequency number | [ 128, 1024, 2048, 4096] |
|  |  | pnums | [0, 1, 10, 15] |
|  |  | lambda | [0, 0.2, 0.4, 0.6, 0.8, 1] |
|  |  | dropout ratio | [0.1, 0.25, 0.5] |
|  |  | feature dimension | 128 |
|  |  | amplitude weight | $0 \sim 1$ |
| Citation |  | epochs | 150 |
|  | C → D | learning rate | [0.001, 0.005, 0.01] |
|  |  | weight decay | [0.001, 0.005, 0.01] |
|  |  | frequency number | [ 128, 1024, 2048, 4096] |
|  |  | pnums | [0, 1, 10, 15] |
|  |  | lambda | [0, 0.2, 0.4, 0.6, 0.8, 1] |

| Dataset | Transfer Scenarios | Hyperparameter | Search Space |
|---|---|---|---|
| | | dropout ratio | [0.1, 0.25, 0.5] |
| | | feature dimension | 128 |
| | | amplitude weight | $0 \sim 1$ |
| | | epochs | 150 |
| | D → A | learning rate | [0.001, 0.005, 0.01] |
| | | weight decay | [0.001, 0.005, 0.01] |
| | | frequency number | [128, 1024, 2048, 4096] |
| | | pnums | [0, 1, 10, 15] |
| | | lambda | [0, 0.2, 0.4, 0.6, 0.8, 1] |
| | | dropout ratio | [0.1, 0.25, 0.5] |
| | | feature dimension | 128 |
| | | amplitude weight | $0 \sim 1$ |
| | | epochs | 150 |
| | D → C | learning rate | [0.001, 0.005, 0.01] |
| | | weight decay | [0.001, 0.005, 0.01] |
| | | frequency number | [ 128, 1024, 2048, 4096] |
| | | pnums | [0, 1, 10, 15] |
| | | lambda | [0, 0.2, 0.4, 0.6, 0.8, 1] |
| | | dropout ratio | [0.1, 0.25, 0.5] |
| | | feature dimension | 128 |
| | | amplitude weight | $0 \sim 1$ |
| | | epochs | 150 |
| | U → B | learning rate | [0.001, 0.005, 0.01] |
| | | weight decay | [0.001, 0.005, 0.01] |
| | | frequency number | [ 128, 1024, 2048, 4096] |
| | | pnums | [0, 1, 5, 10, 15] |
| | | lambda | [0, 0.2, 0.4, 0.6, 0.8, 1] |
| | | dropout ratio | [0.1, 0.25, 0.5] |
| | | feature dimension | 128 |
| | | amplitude weight | $0 \sim 1$ |
| | | epochs | 150 |
| | U→ E | learning rate | [0.001, 0.005, 0.01] |
| | | weight decay | [0.001, 0.005, 0.01] |
| | | frequency number | [32, 64, 128, 1024, 2048, 4096] |
| | | pnums | [0, 1, 5, 10, 15] |
| | | lambda | [0, 0.2, 0.4, 0.6, 0.8, 1] |
| | | dropout ratio | [0.1, 0.2, 0.25, 0.5] |
| | | feature dimension | 128 |
| | | amplitude weight | $0 \sim 1$ |
| | | epochs | 150 |
| | B → U | learning rate | [0.001, 0.005, 0.01] |
| | | weight decay | [0.001, 0.005, 0.01] |
| | | frequency number | [32, 64, 128, 1024, 2048, 4096] |
| | | pnums | [0, 1, 5, 10, 15] |
| | | lambda | [0, 0.2, 0.4, 0.6, 0.8, 1] |
| | | dropout ratio | [0.1, 0.25, 0.5] |
| | | feature dimension | 128 |
| | | amplitude weight | $0 \sim 1$ |
| Airport | | epochs | 150 |
| | B → E | learning rate | [0.001, 0.003, 0.005, 0.01] |
| | | weight decay | [0.001, 0.005, 0.01] |
| | | frequency number | [32, 64, 128, 1024, 2048, 4096] |
| | | pnums | [0, 1, 5, 10, 15] |
| | | lambda | [0, 0.2, 0.4, 0.6, 0.8, 1] |
| | | dropout ratio | [0.1, 0.25, 0.5] |
| | | feature dimension | 128 |
| | | amplitude weight | $0 \sim 1$ |
| | | epochs | 150 |

| Dataset | Transfer Scenarios | Hyperparameter | Search Space |
|---|---|---|---|
| | E → U | learning rate | [0.001, 0.005, 0.01] |
| | | weight decay | [0.001, 0.005, 0.01] |
| | | frequency number | [32, 64, 128, 1024, 2048, 4096] |
| | | pnums | [0, 1, 10, 15] |
| | | lambda | [0, 0.2, 0.4, 0.6, 0.8, 1] |
| | | dropout ratio | [0.1, 0.25, 0.5] |
| | | feature dimension | 128 |
| | | amplitude weight | $0 \sim 1$ |
| | | epochs | 150 |
| | E → B | learning rate | [0.001, 0.003, 0.005, 0.01] |
| | | weight decay | [0.001, 0.005, 0.01] |
| | | frequency number | [32, 64, 128, 1024, 2048, 4096] |
| | | pnums | [0, 1, 10, 15] |
| | | lambda | [0, 0.2, 0.4, 0.6, 0.8, 1] |
| | | dropout ratio | [0.1, 0.25, 0.5] |
| | | feature dimension | 128 |
| | | amplitude weight | $0 \sim 1$ |
| | | epochs | 150 |
| Blog | B1 → B2 | learning rate | [0.001, 0.005, 0.01] |
| | | weight decay | [0.001, 0.005, 0.01] |
| | | frequency number | [32, 64, 128, 1024, 2048, 4096] |
| | | pnums | [0, 1, 10, 15] |
| | | lambda | [0, 0.2, 0.4, 0.6, 0.8, 1] |
| | | dropout ratio | [0.1, 0.25, 0.5] |
| | | feature dimension | 128 |
| | | amplitude weight | $0 \sim 1$ |
| | | epochs | 150 |
| | B2 → B1 | learning rate | [0.001, 0.005, 0.01] |
| | | weight decay | [0.001, 0.005, 0.01] |
| | | frequency number | [32, 64, 128, 1024, 2048, 4096] |
| | | pnums | [0, 1, 10, 15] |
| | | lambda | [0, 0.2, 0.4, 0.6, 0.8, 1] |
| | | dropout ratio | [0.1, 0.25, 0.5] |
| | | feature dimension | 128 |
| | | amplitude weight | $0 \sim 1$ |
| | | epochs | 150 |
| Twitch | DE → EN | learning rate | [0.001, 0.005, 0.01] |
| | | weight decay | [0.001, 0.005, 0.01] |
| | | frequency number | [32, 64, 128, 1024, 2048, 4096] |
| | | pnums | [0, 1, 10, 15] |
| | | lambda | [0, 0.2, 0.4, 0.6, 0.8, 1] |
| | | dropout ratio | [0.1, 0.25, 0.5] |
| | | feature dimension | 128 |
| | | amplitude weight | $0 \sim 1$ |
| | | epochs | 150 |
| | EN → DE | learning rate | [0.001, 0.005, 0.01] |
| | | weight decay | [0.001, 0.005, 0.01] |
| | | frequency number | [32, 64, 128, 1024, 2048, 4096] |
| | | pnums | [0, 1, 10, 15] |
| | | lambda | [0, 0.2, 0.4, 0.6, 0.8, 1] |
| | | dropout ratio | [0.1, 0.25, 0.5] |
| | | feature dimension | 128 |
| | | amplitude weight | $0 \sim 1$ |
| | | epochs | 150 |

Table 7: Parameter search space list.

