# OpenReview forum: "Learning Adaptive Distribution Alignment with Neural Characteristic Function for Graph Domain Adaptation"
_ICLR.cc/2026/Conference — ICLR 2026 Poster_

### Official Review · Reviewer_TwjK · 2025-10-27

**Soundness:** 3
**Presentation:** 3
**Contribution:** 3
**Rating:** 6
**Confidence:** 5

**Summary:**

This paper introduces ADAlign, an adaptive distribution alignment framework for graph domain adaptation (GDA). The authors propose to address the challenge of complex, multi-faceted distributional shifts between source and target graphs by leveraging neural characteristic functions (CF) in the spectral domain. The key contribution is the Neural Spectral Discrepancy (NSD), which uses characteristic functions to capture feature-structure dependencies and employs a learnable frequency sampler to adaptively identify discriminative spectral components through minimax optimization. The framework is evaluated on 10 datasets across 16 transfer tasks, demonstrating superior performance compared to state-of-the-art baselines.

**Strengths:**

1. **Clear Problem Formulation**: The paper effectively motivates the need for adaptive alignment by showing how dominant discrepancy sources vary across transfer scenarios (Figure 1), making a compelling case against fixed heuristic alignment strategies.

2. **Theoretical Grounding**: The use of characteristic functions is theoretically well-motivated, with proper mathematical formulation including convergence and uniqueness theorems. The PAC-Bayesian analysis in Section 4.5 provides theoretical justification for the approach.

3. **Comprehensive Experiments**: The experimental evaluation is extensive, covering 16 transfer scenarios across diverse graph domains (citation networks, airports, social networks) with consistent improvements over 15 baseline methods.

**Weaknesses:**

1. The paper fails to clearly distinguish its characteristic function-based approach from existing spectral alignment methods in GDA. Most critically, the relationship to SA-GDA (Pang et al., 2023), which also performs spectral augmentation for graph domain adaptation, is not discussed. SA-GDA already demonstrated that spectral domain alignment can effectively handle distribution shifts in graphs by augmenting the graph Laplacian spectrum.

2. The paper emphasizes the amplitude-phase decomposition (Equation 7) but doesn't convincingly demonstrate why this complex-valued representation is necessary.

3. The adaptive frequency sampler (Section 4.3) appears to be a standard importance sampling approach: The normal scale mixture formulation (Equation 10) is not well justified - why this specific parametrization? Minimax optimization can lead to instability, though the authors claim stability without showing the training dynamics of the frequency sampler itself—missing comparisons with simple fixed-frequency grids or other sampling strategies beyond the limited ablation in Section 5.4.

4.  The baselines don't include recent spectral methods like SA-GDA (Pang et al., 2023) or other Fourier-based domain adaptation approaches. The significance tests (Appendix E.2) compare against only three baselines, not the full set. The visualization in Figure 10 is not particularly convincing - the claimed "sharp boundaries" in ADAlign's embeddings are not clearly superior to baselines.

**Questions:**

See Weaknesses.

---

> ### Author Response · Authors · 2025-11-21
> **Response to TwjK (1/2)**
>
> We are grateful for your efforts in reviewing the paper. Below, we do our utmost to address your concerns.
>
> >**Q1: The relationship to SA-GDA (Pang et al., 2023) is not discussed. And the baselines don't include recent spectral methods like SA-GDA or other Fourier-based domain adaptation approaches.**
>
> Thanks!
>
>  Firstly, although SA-GDA also works in the spectral domain, its formulation is different from ours. SA-GDA assumes that nodes belonging to the same class share similar spectral patterns across domains, and it manually constructs low-/high-pass filters to inject target spectral components into source features for class-wise alignment.
>
> Secondly, ADAlign does not rely on predefined rules. Instead, it represents distributions through neural characteristic functions and learns which frequencies are most informative for each transfer task. This allows ADAlign to model composite and scenario-dependent shifts in a unified and fully data-driven manner, without requiring a fixed spectral pipeline.
>
> Finally, following your suggestion, the comparative results between SA-GDA and our method across multiple transfer tasks are summarized below:
>
> ---- **Table 1: Results between SA-GDA and our method** ----
> | Method | Metric | A→C | A→D | C→A | C→D | D→A | D→C |
> |--------|--------|-----|-----|-----|-----|-----|-----|
> | SA-GDA (MM'23) | Mi-F1 | 76.02 ± 1.57 | 70.36 ± 1.08 | 70.84 ± 1.87 | 75.48 ± 0.20 | 63.41 ± 1.12 | 72.81 ± 1.66 |
> | Ours | Mi-F1 | **82.21 ± 0.36** | **78.06 ± 0.61** | **76.24 ± 0.29** | **79.51 ± 0.20** | **74.38 ± 0.98** | **81.84 ± 0.10** |
>
> | Method | Metric | U→B | U→E | B→U | B→E | E→U | E→B |
> |--------|--------|-----|-----|-----|-----|-----|-----|
> | SA-GDA (MM'23) | Ma-F1 | 67.60 ± 1.54 | 51.09 ± 1.02 | 48.20 ± 3.29 | 56.99 ± 0.32 | 47.78 ± 2.42 | 63.53 ± 1.82 |
> | Ours | Ma-F1 | **72.52 ± 0.41** | **53.23 ± 0.68** | **55.71 ± 0.59** | **58.49 ± 1.19** | **55.41 ± 0.29** | **75.82 ± 0.47** |
>
> As shown in the tables, our method consistently outperforms SA-GDA across all transfer tasks, demonstrating the effectiveness of our proposed approach.
>
> >**Q2: The paper doesn't convincingly demonstrate why this complex-valued representation is necessary.**
>
>  Thank you for the comment!
>
> The amplitude-phase decomposition is introduced not merely for mathematical convenience, but to enable finer control over different types of spectral shifts. The amplitude term aligns the amount of energy carried by each frequency component, which reflects global and smooth structural patterns, while the phase term aligns how these components are arranged and correlated, capturing finer relational or heterophilic irregularities. Separating these two aspects allows ADAlign to weight and align them differently when needed, resulting in a more nuanced and effective distribution alignment than using magnitude alone.

---

> > ### Author Response · Authors · 2025-11-21
> > **Response to TwjK (2/2)**
> >
> > >**Q3: The normal scale mixture formulation (Equation 10) is not well justified - why this specific parametrization? And missing comparisons with simple fixed-frequency grids or other sampling strategies beyond the limited ablation in Section 5.4.**
> >
> > Thanks for the insightful comment!
> >
> > Firstly, our motivation for using a learnable normal scale mixture is to overcome the limitations of non-adaptive sampling schemes. Fixed-frequency grids or single Gaussian samplers implicitly assume one spectral structure fits all tasks, yet different transfers exhibit very different spectral discrepancies. Some are dominated by smooth low-frequency shifts, while others require sharper high-frequency components. A single Gaussian or a manually specified finite mixture enforces a fixed spectral bias and cannot flexibly capture both patterns.
> >
> > Secondly, the normal scale mixture offers a simple yet flexible parametrization whose mixing distribution adapts its tail behavior and effective bandwidth to the data. This enables the sampler to automatically shift probability mass between low- and high-frequency regions, rather than relying on a fixed, hand-designed frequency set.
> >
> > Finally, to address the your suggestion, we compared our learnable scale-mixture sampler with two non-parametric alternatives: a single fixed Gaussian sampler, and a fixed two-component Gaussian mixture. Both remove the parametrization in Eq. 10 and represent non-adaptive frequency selection. Our parametrized scale mixture consistently achieves better and more stable alignment performance:
> >
> >
> > ---- **Table 2: Results on different sampler type** ----
> >
> > | Sampler Type                | A→C (Mi-F1)     | A→C (Ma-F1)     | A→D (Mi-F1)     | A→D (Ma-F1)     |
> > |-----------------------------|------------------|------------------|------------------|------------------|
> > | Single Gaussian             | 75.45 ± 0.92     | 72.20 ± 1.30     | 71.38 ± 0.22     | 67.80 ± 0.90     |
> > | Fixed Gaussian Mixture      | 76.14 ± 0.75     | 74.80 ± 0.90     | 71.24 ± 0.64     | 67.50 ± 1.15     |
> > | Normal Scale Mixture (ours) | **78.25 ± 0.37** | **76.12 ± 0.45** | **75.12 ± 0.51** | **72.53 ± 1.29** |
> >
> >
> >
> >
> > >**Q4: The significance tests (Appendix E.2) compare against only three baselines, not the full set.**
> >
> > Thank you for pointing this out!
> >
> > Firstly, we initially reported significance tests only against three baselines (A2GNN, GAA, HGDA) because they are the most representative recent GDA methods in our setting.
> >
> > Then, following your suggestion, we have now extended the analysis and included significance test results for all models **in Figure 8** and **in Figure 9**  of the revised version, making the statistical comparison complete and transparent.
> >
> > >**Q5: The visualization in Figure 10 is not particularly convincing - the claimed "sharp boundaries" in ADAlign's embeddings are not clearly superior to baselines.**
> >
> > Thanks！
> >
> > we have updated corresponding visualization **in Figure 7** in the revised version to make the comparison clearer.

---

> > > ### Comment · Reviewer_TwjK · 2025-11-26
> > >
> > > Dear Authors,
> > >
> > > I thank the authors for their detailed response.
> > >
> > > I have also carefully read the comments from the other reviewers. At this moment, I will maintain my current rating and make a final decision after a discussion with other reviewers and AC.
> > >
> > > Reviewer TwjK

---

> > > > ### Author Response · Authors · 2025-11-27
> > > > **Thanks for positive feedback**
> > > >
> > > > Dear Reviewer TwjK,
> > > >
> > > > We sincerely appreciate your valuable reviews and positive feedback. We hope the idea and approach presented in this work can inspire more studies in this direction.
> > > >
> > > > If you have any questions, please feel free to discuss them with us at any time.
> > > >
> > > > Sincerely,
> > > >
> > > > Authors of Submission 11024

---

### Official Review · Reviewer_HQfr · 2025-10-30

**Soundness:** 4
**Presentation:** 4
**Contribution:** 3
**Rating:** 8
**Confidence:** 2

**Summary:**

This paper introduces ADAlign, a framework for graph domain adaptation (GDA) that addresses complex distributional shifts between source and target graphs. It proposes the neural spectral discrepancy (NSD), which measures cross-graph gaps in the spectral domain using neural characteristic functions, and adaptively samples informative frequencies via a minimax optimization. The method is theoretically supported by PAC-Bayesian analysis and achieves state-of-the-art performance across 16 transfer scenarios with lower computational and memory costs.

**Strengths:**

1. Figure 1 clearly illustrates the limitations of existing approaches, providing strong motivation for introducing adaptive alignment.
2. The introduction of NSD based on neural characteristic functions is theoretically grounded. By performing adaptive alignment in the spectral domain, the method provides a more flexible framework.
3. The experiments are comprehensive, covering 13 strong baselines across 16 transfer scenarios, with ablation studies and parameter sensitivity analyses supporting the method’s validity.
4. ADAlign achieves significant improvements in computational efficiency, being faster and using less memory than baselines.

**Weaknesses:**

1. Although the paper explains that low-frequency captures large-scale structures and high-frequency captures fine-grained structures, it does not clearly clarify what specific graph structural characteristics the learned frequency distributions reflect.
2. There is no analysis or experiment showing how replacing the GNN backbone affects the NSD-based alignment performance.
3. While the paper provides a complexity analysis, the paper lacks experiments on the scalability of ADAlign to very large graphs or how its performance changes with graph size.

**Questions:**

Look at the Weaknesses.

---

> ### Author Response · Authors · 2025-11-21
> **Response to HQfr (1/2)**
>
> Thank you for your recognition on our work! Below, we do our utmost to address your concerns.
>
>
> >**Q1: It does not clearly clarify what specific graph structural characteristics the learned frequency distributions reflect.**
>
> Thank you for the suggestion!
>
> Firstly, in graph signal processing, the spectrum of a graph (e.g., Laplacian eigenvalues and eigenvectors) provides a frequency view of structure: (1) Low frequencies correspond to smooth variations over the graph and are closely related to global communities, large-scale connectivity patterns, and homophilous regions. (2) High frequencies correspond to rapid changes across edges and are associated with local irregularities, such as hubs vs. periphery, bridge or outlier nodes, and heterophilous or noisy edges that break community consistency.
>
> Secondly, our method is designed from this spectral perspective: we aim to align domains by matching how information is distributed across these low‑ and high‑frequency components. However, ADAlign operates on GNN embeddings rather than directly on the raw adjacency or Laplacian spectrum. This means we do not hand‑select specific handcrafted structural statistics. Instead, the model learns, for each scenario, an adaptive frequency distribution over the embedding space that best captures domain‑invariant structural patterns.
>
> Although the encoder fuses structure with attributes and compresses them, the learned frequency distributions remain interpretable at this coarse level: the low‑frequency mass reflects alignment of global, smooth structural patterns (communities, degree/connectivity trends), while the high‑frequency mass reflects alignment of more local, irregular structures (hubs, bridges, heterophilous or noisy edges), which is consistent with standard interpretations of graph Fourier modes.
>
>
> >**Q2: There is no analysis or experiment showing how replacing the GNN backbone affects the NSD-based alignment performance..**
>
> Thanks!
>
> To further verify robustness, we instantiated ADAlign on two standard GNN backbones, GCN and GAT, and compared it with strong baselines on two transfer scenarios. The results are shown below:
>
>
> ---- **Table 1: Results on the GCN backbone** ----
>
>
> | Method          | A→C (Mi-F1)     | A→C (Ma-F1)     | A→D (Mi-F1)     | A→D (Ma-F1)     |
> |-----------------|---------------------|---------------------|---------------------|---------------------|
> | GCN (ICLR'17)   | 70.82 ± 1.26        | 66.49 ± 2.21        | 65.05 ± 2.15        | 59.53 ± 0.44        |
> | GAA (ICLR'25)   | 73.50 ± 0.85        | 70.20 ± 1.30        | 67.30 ± 1.50        | 62.80 ± 0.90        |
> | HGDA (ICML'25)  | 75.40 ± 0.60        | 72.80 ± 0.90        | 68.65 ± 1.10        | 64.50 ± 1.15        |
> | ADAlign (Ours)  | **76.99 ± 0.37**        | **75.11 ± 0.45**        | **70.11 ± 0.80**        | **66.33 ± 1.33**        |
>
>
> ---- **Table 2: Results on the GAT backbone** ----
>
> | Method          | A→C (Mi-F1)     | A→C (Ma-F1)     | A→D (Mi-F1)     | A→D (Ma-F1)     |
> |-----------------|---------------------|---------------------|---------------------|---------------------|
> | GAT (ICLR'18)   | 62.77 ± 2.24        | 59.60 ± 2.82        | 60.29 ± 1.33        | 55.09 ± 1.16        |
> | GAA (ICLR'25)   | 67.63 ± 1.42        | 64.53 ± 1.74        | 63.33 ± 1.03        | 59.26 ± 2.06        |
> | HGDA (ICML'25)  | 65.20 ± 1.80        | 62.10 ± 2.20        | 61.80 ± 1.20        | 57.30 ± 1.60        |
> | ADAlign (Ours)  | **70.45 ± 0.95**        | **69.80 ± 1.10**        | **67.45 ± 0.85**        | **65.90 ± 1.25**        |
>
>
> Across both backbones, ADAlign consistently outperform the corresponding source-only models and all competing baselines, often by a clear margin. These results confirm that ADAlign is not tied to a specific encoder and can be reliably integrated with different GNN architectures.
>
> In the revised version, we have included these results **on page 20**.

---

> > ### Author Response · Authors · 2025-11-21
> > **Response to HQfr (2/2)**
> >
> > >**Q3: The paper lacks experiments on the scalability of ADAlign to very large graphs.**
> >
> > Thanks!
> >
> >  To assess the model’s generalization to substantially larger graphs, we additionally evaluated ADAlign on two popular subsets of the Microsoft Academic Graph. These graphs are an order of magnitude larger than those in current experiments (e.g., ACMv9: 9,360 nodes, 31,112 edges, 5 labels).
> > The dataset statistics are:
> >
> > ---- **Table 3: Large-scale dataset statistics** ----
> >
> > | Dataset   | #Nodes   | #Edges   | #Labels |
> > |-----------|----------|----------|---------|
> > | China (CN) | 101,952  | 285,991  | 20      |
> > | USA (US)   | 132,558  | 702,482  | 20      |
> >
> > We compared ADAlign with the competitive GDA baselines under two transfer settings (CN→US and US→CN). The Micro-/Macro-F1 scores are:
> >
> > ---- **Table 4: Performance on large-scale datasets** ----
> >
> > | Method          | CN→US (Mi-F1)     | CN→US (Ma-F1)     | US→CN (Mi-F1)     | US→CN (Ma-F1)     |
> > |-----------------|---------------------|---------------------|---------------------|---------------------|
> > | DGSDA (ICML'25)           | 46.97 ± 0.54        | 10.39 ± 0.21        | 60.26 ± 0.37        | 10.17 ± 1.12        |
> > | GAA (ICLR'25)             | 47.17 ± 0.49        | 47.17 ± 0.49        | OOM                 | OOM                 |
> > | HGDA (ICML'25)            | OOM                 | OOM                 | OOM                 | OOM                 |
> > | ADAlign (Ours)  | **52.29 ± 0.22**    | **49.56 ± 1.07**         | **62.78 ± 0.06**    | **13.47 ± 0.85**    |
> >
> > ADAlign achieves the best performance on both tasks and remains trainable without out-of-memory (OOM) issues, whereas several strong baselines (GAA, HGDA) fail on at least one task. These results support that our spectral alignment framework scales to much larger graphs in practice.

---

### Official Review · Reviewer_2AFp · 2025-11-01

**Soundness:** 3
**Presentation:** 3
**Contribution:** 3
**Rating:** 6
**Confidence:** 3

**Summary:**

This paper proposes ADAlign, an adaptive distribution alignment framework for Graph Domain Adaptation to address complex distributional shifts between source and target graphs. It introduces Neural Spectral Discrepancy (NSD), a parametric distance leveraging neural characteristic functions in the spectral domain to capture multi-level feature-structure dependencies. Equipped with a learnable frequency sampler and minimax optimization, ADAlign automatically identifies and aligns discrepancies without manual criteria.

**Strengths:**

1. Eliminates manual specification of alignment criteria by dynamically prioritizing relevant spectral components for each transfer scenario.
2. NSD integrates amplitude and phase differences for a comprehensive view of cross-graph shifts.
3. Performs consistently well across domains.

**Weaknesses:**

1. Regarding the acceleration aspect, I understand the paper adopts the spectral method. However, this approach does not have low computational complexity. What methods are used for acceleration or approximation in the paper? It would be better if the authors could provide a proof of the complexity.
2. The perspective proposed in the paper is good, but I am not very familiar with the statement in the introduction: "these approaches often rely on heuristic strategies that first manually design graph filters to extract relevant features" and feel that it is not the mainstream method in Graph Domain Adaptation. Proposing a model based on this point does not seem to be an appealing argument. Besides, it would also be helpful if the paper provides some descriptions of this pipeline to assist readers' understanding.
3. The KL divergence in Figure 1 is used to measure the difference between features. Can this be regarded as assuming that the dataset satisfies Ps(Y|X) = Pt(Y|X)? Otherwise, merely exploring the difference between Ps(X) and Pt(X) seems irrelevant to transfer learning.
4. There is a lack of experiments on large datasets. All datasets used here are too small, such as Brazil.

**Questions:**

See Weaknesses

---

> ### Author Response · Authors · 2025-11-21
> **Response to 2AFp (1/2)**
>
> Thanks for your encouraging and valuable comments! Below, we respond to your questions in detail.
> >**Q1: Regarding the acceleration aspect, I understand the paper adopts the spectral method. However, this approach does not have low computational complexity. What methods are used for acceleration or approximation in the paper? It would be better if the authors could provide a proof of the complexity.**
>
> Thanks! Your understanding is correct: if we were to align all frequencies, a purely spectral approach would indeed be expensive. Our efficiency mainly comes from two design choices.
>
> Firstly, rather than integrating over the entire spectral domain, we sample only a small number of frequencies and approximate NSD via Monte Carlo. The cost of evaluating all characteristic functions is $\mathcal{O}(M (N_\mathsf{S} + N_\mathsf{T}) d)$, where $M$ is the number of sampled frequencies and $d$ is the embedding dimension, $N_\mathsf{S}$ and $N_\mathsf{T}$ are the numbers of source and target nodes in the batch. In practice, using up to 2048 frequencies is sufficient, and both performance and runtime saturate beyond this point (see Figure 6). This keeps the computation compact while still capturing the informative spectral regions.
>
>
> Secondly, ADAlign does not require eigen-decomposition or large kernel matrices (e.g., those used in MMD-based alignment). All computations reduce to inner products between embeddings and sampled frequencies, which are inexpensive and highly parallelizable.
>
> >**Q2: I am not very familiar with the statement in the introduction: "these approaches often rely on heuristic strategies that first manually design graph filters to extract relevant features" and feel that it is not the mainstream method in Graph Domain Adaptation. Proposing a model based on this point does not seem to be an appealing argument. Besides, it would also be helpful if the paper provides some descriptions of this pipeline to assist readers' understanding.**
>
> Thanks for the comment!
>
> Firstly, our intention was not to suggest that manually designed graph filters represent the mainstream in GDA, most early and concurrent GDA methods indeed rely on global embedding alignment with generic distances (KL, MMD, Wasserstein, etc.), and our work follows this general direction.
>
> Secondly, what we aimed to clarify is that a recent line of fine-grained GDA approaches explicitly decomposes distribution shifts and introduces hand-crafted structural or feature extractors before alignment. For example, GRADE builds WL-subtree–based filters, GAA introduces an attention-style cross-channel module over pre-selected attribute groups, and HGDA designs mixed homophily/heterophily filters to capture relational patterns. These methods are technically sophisticated and published at leading venues, but they all require the designer to pre-specify which properties (subtrees, homophily, channels, etc.) should be extracted and aligned, each with its own heuristic metric.
>
> Finally, such manually selected filters may become brittle when the dominant discrepancies vary across scenarios (as shown in Figure 1). Our contribution offers a complementary perspective: instead of committing to a fixed set of graph filters and alignment metrics, we rely on neural characteristic functions and an adaptive frequency sampler that automatically identifies and emphasizes the spectral components most informative for each transfer task.
>
>
> >**Q3: Can this be regarded as assuming that the dataset satisfies $
> P_s(Y \mid X) = P_t(Y \mid X)
> $?**
>
> Yes. This assumption is widely adopted in many prior GDA studies [1,2,3,4,5], and our work follows the same setting. These methods typically operate under the covariate shift scenario, where the input distribution differs across domains while the conditional label distribution remains invariant. This allows the model to focus on aligning feature distributions or structural properties across domains without altering the underlying labeling mechanism.
>
> **Reference**
>
> [1] Structural re-weighting improves graph domain adaptation, ICML, 2024.
>
> [2] Revisiting, benchmarking and understanding unsupervised graph domain adaptation, NeurIPS, 2024.
>
> [3] Can Modifying Data Address Graph Domain Adaptation? KDD, 2024.
>
> [4] A simple yet effective approach for unsupervised graph domain adaptation, AAAI, 2025.
>
> [5] On the benefits of attribute-driven graph domain adaptation, ICLR, 2025.

---

> > ### Author Response · Authors · 2025-11-21
> > **Response to 2AFp (2/2)**
> >
> > >**Q4: There is a lack of experiments on large datasets.**
> >
> > Thanks!
> >
> >  To assess the model’s generalization to substantially larger graphs, we additionally evaluated ADAlign on two popular subsets of the Microsoft Academic Graph. These graphs are an order of magnitude larger than those in current datasets (e.g., ACMv9: 9,360 nodes, 31,112 edges, 5 labels).
> > The dataset statistics are:
> >
> > ---- **Table 1: Large-scale dataset statistics** ----
> >
> > | Dataset   | #Nodes   | #Edges   | #Labels |
> > |-----------|----------|----------|---------|
> > | China (CN) | 101,952  | 285,991  | 20      |
> > | USA (US)   | 132,558  | 702,482  | 15      |
> >
> > We compared ADAlign with the competitive GDA baselines under two transfer settings (CN→US and US→CN). The Micro-/Macro-F1 scores are:
> >
> > ---- **Table 2: Performance on large-scale datasets** ----
> >
> > | Method          | CN→US (Mi-F1)     | CN→US (Ma-F1)     | US→CN (Mi-F1)     | US→CN (Ma-F1)     |
> > |-----------------|---------------------|---------------------|---------------------|---------------------|
> > | DGSDA (ICML'25)           | 46.97 ± 0.54        | 10.39 ± 0.21        | 60.26 ± 0.37        | 10.17 ± 1.12        |
> > | GAA (ICLR'25)             | 47.17 ± 0.49        | 47.17 ± 0.49        | OOM                 | OOM                 |
> > | HGDA (ICML'25)            | OOM                 | OOM                 | OOM                 | OOM                 |
> > | ADAlign (Ours)  | **52.29 ± 0.22**    | **49.56 ± 1.07**         | **62.78 ± 0.06**    | **13.47 ± 0.85**    |
> >
> > ADAlign achieves the best performance on both tasks and remains trainable without out-of-memory (OOM) issues, whereas several strong baselines (GAA, HGDA) fail on at least one task. These results support that our spectral alignment framework scales to much larger graphs in practice.

---

> > > ### Comment · Reviewer_2AFp · 2025-11-27
> > > **Thank you**
> > >
> > > Thanks for the detailed response. I would keep my positive score.

---

### Official Review · Reviewer_A6vr · 2025-11-01

**Soundness:** 3
**Presentation:** 3
**Contribution:** 3
**Rating:** 6
**Confidence:** 2

**Summary:**

This paper introduced the ADAlign framework, a novel approach to handling composite
distribution shifts in GDA. By leveraging characteristic function in the complex Fourier domain, ADAlign dynamically identifies and aligns discriminative spectral components through minimax optimization, eliminating the need for manual reweighting or heuristic metrics. Extensive experiments across 16 benchmarks show that ADAlign consistently outperforms state-of-the-art methods, while also improving training stability and reducing computational overhead. These advantages make ADAlign a highly effective and practical solution for real-world graph transfer learning tasks.

**Strengths:**

1. Novel methodology: This paper introduces NSD, a novel parametric distance
for graphs that leverages neural characteristic function in the spectral domain to capture multi-level feature-structure dependencies, providing a unified approach to quantifying distributional shifts.
2. Effective adaptive mechanism: This paper proposes an adaptive framework that automatically identifies and aligns the most relevant sources of discrepancy in each transfer scenario, enabling flexible and task-aware adaptation to dynamic graph shifts.
3. Exceptional Empirical Performance and Significant Computational Efficiency: Extensive experiments in this paper show that ADAlign outperforms state-of-the-art baselines, significantly reducing memory consumption and training time.

**Weaknesses:**

1. The paper's core philosophy is adaptivity, yet the amplitude weight k, which balances amplitude and phase, is a hand-tuned hyperparameter. The analysis in Figure 6 shows an optimal range, but it's possible that it is also scenario-dependent. A fixed k seems slightly at odds with the "adaptive" goal.
2. It would be helpful if the paper could further justify modeling the sampler as a normal scale mixture, and clarify whether simpler parameterizations such as a single Gaussian, a fixed finite mixture would already work in practice.
3. The paper adopts the GNN encoder  F (Liu et al., 2024a) to map each graph into node-level embeddings,  but it remains unclear how sensitive ADAlign is to the choice of backbone such as GCN and GAT.
4. There is a minor inconsistency in the wording. For example, the main text states ‘We adopt the GNN encoder F (Liu et al., 2024a),’ whereas Figure 2 describes the components as ‘each processed by a GCN.

**Questions:**

1. Regarding Weakness 1: Did the authors consider making the amplitude weight k adaptive?
2. If you replace the GNN backbone with GCN or GAT, will the advantages of your model be offset?
3. Could the authors visualize the learned frequency sampling distributions across different datasets?

---

> ### Author Response · Authors · 2025-11-21
> **Response to A6vr (1/2)**
>
> Thanks a lot for your efforts in reviewing our work.
>
> >**Q1: A fixed $\kappa$ seems slightly at odds with the "adaptive" goal. Did the authors consider making the amplitude weight $\kappa$ adaptive?**
>
> Thank you for this thoughtful comment！
>
>  Firstly, the “adaptivity’’ in ADAlign mainly concerns **where** to align in the spectral domain (Which frequencies). Within each sampled frequency, we decompose the discrepancy into amplitude and phase terms and combine them with a single scalar weight $\kappa$. In our main experiments, $\kappa$ is fixed globally, and Figure 6 shows that ADAlign is robust across a wide range (0.65–0.75), indicating that the primary adaptive behavior comes from the sampling mechanism rather than sensitivity to the exact value of $\kappa$.
>
> Seconfly, in our framework, $\kappa$ **can also be easily extended** to an adaptive form, and we have explored this variant. For a given batch and sampled frequencies ${\mathbf{t}_m}$, we compute the average amplitude and phase discrepancies:
> $$
> \text{amp} = E_t\big( (\Psi_S(t) - \Psi_T(t))^2 \big), \quad
> \text{pha} = E_t\big( 2\, \Psi_S(t)\, \Psi_T(t)\, (1 - \cos(\theta_S(t) - \theta_T(t))) \big).
> $$
> We then define an instantaneous, data-dependent weight: $
> \kappa = \frac{\text{amp}}{\text{amp} + \text{pha} + \varepsilon },
> $ where $\varepsilon$ avoids division by zero. This behaves as expected: when amplitude differences dominate, $\kappa$ increases and places more emphasis on amplitude. When phase discrepancies dominate, $\kappa$ decreases accordingly. Thus $\kappa$ becomes scenario- and iteration-dependent while remaining numerically stable.
>
> Finally, to evaluate this adaptive design, we tested the adaptive-$\kappa$ variant on two transfer scenarios (A→C and A→D):
>
> ---- **Table 1: Results on the adaptive-$\kappa$ variant** ----
>
> | Method                   | A→C (Mi-F1)       | A→C (Ma-F1)       | A→D (Mi-F1)       | A→D (Ma-F1)       |
> |--------------------------|--------------------|--------------------|--------------------|--------------------|
> | ADAlign (fixed-κ)        | **78.25 ± 0.37**       | 76.12 ± 0.45      | 74.12 ± 0.51       | 72.53 ± 1.29       |
> | ADAlign (adaptive-κ)     | 78.11 ± 0.21       | **77.25 ± 0.42**       | **75.25 ± 0.15**       | **73.42 ± 1.07**       |
>
>
> In both cases, the adaptive version provided a favorable performance advantage over the best fixed-$\kappa$ setting in most scenarios.
>
> Following your suggestion, we will consider incorporating this adaptive $\kappa$ strategy in future work.
>
>
> >**Q2: It would be helpful if the paper could further justify modeling the sampler as a normal scale mixture, and clarify whether simpler parameterizations such as a single Gaussian, a fixed finite mixture would already work in practice.**
>
> Thanks for the insightful comment!
>
> Firstly, our choice of a normal scale mixture is motivated by prior generative modeling work [1], where flexible latent priors such as scale mixtures can model complex distributions while remaining easy to sample from. In our setting, **the sampler must adapt to very different spectral discrepancy patterns** across tasks—some dominated by smooth low-frequency shifts, others by sharp high-frequency changes. A single Gaussian or a fixed finite mixture imposes a rigid bias, either concentrating too narrowly on one region or spreading too uniformly across all frequencies, making it difficult for these simpler forms to capture both coarse and fine-grained discrepancies.
>
> By contrast, a normal scale mixture with a learnable mixing distribution provides a **compact yet highly expressive** family that subsumes a wide range of tails and shapes. During the minimax optimization, the mixing distribution adapts its shape to emphasize the relevant parts of the spectrum for each task. This flexibility allows the sampler to move smoothly between low- and high-frequency regions without being restricted to a fixed set of components.
>
> Secondly, to verify this design, we compared three parameterizations on two  transfer scenarios: a single Gaussian sampler, a fixed two-component Gaussian mixture, and our normal scale mixture. The normal scale mixture consistently provides more stable performance across tasks. The results are shown below.
>
> ---- **Table 2: Results on different sampler type** ----
>
> | Sampler Type                | A→C (Mi-F1)     | A→C (Ma-F1)     | A→D (Mi-F1)     | A→D (Ma-F1)     |
> |-----------------------------|------------------|------------------|------------------|------------------|
> | Single Gaussian             | 75.45 ± 0.92     | 72.20 ± 1.30     | 71.38 ± 0.22     | 67.80 ± 0.90     |
> | Fixed Gaussian Mixture      | 76.14 ± 0.75     | 74.80 ± 0.90     | 71.24 ± 0.64     | 67.50 ± 1.15     |
> | Normal Scale Mixture (ours) | **78.25 ± 0.37** | **76.12 ± 0.45** | **75.12 ± 0.51** | **72.53 ± 1.29** |
>
> Reference
>
> [1] A characteristic function approach to deep implicit generative modeling, CVPR, 2020.

---

> > ### Author Response · Authors · 2025-11-21
> > **Response to A6vr (2/2)**
> >
> > >**Q3: How sensitive is the proposed method to the choice of GNN backbone, such as GCN or GAT?**
> >
> >
> > Thank you for the question！
> >
> >
> > Firstly, in the main experiments we use GNN encoder from Liu et al. (2024a) because it is the current state-of-the-art GDA backbone. Using a common encoder for all approaches ensures a fair comparison and avoids conflating backbone effects with alignment effects.
> >
> > Secondly, to further verify robustness, we instantiated ADAlign on two standard GNN backbones, GCN and GAT, and compared it with strong baselines on two transfer scenarios. The results are shown below:
> >
> > ---- **Table 3: Results on the GCN backbone** ----
> >
> >
> > | Method          | A→C (Mi-F1)     | A→C (Ma-F1)     | A→D (Mi-F1)     | A→D (Ma-F1)     |
> > |-----------------|---------------------|---------------------|---------------------|---------------------|
> > | GCN (ICLR'17)   | 70.82 ± 1.26        | 66.49 ± 2.21        | 65.05 ± 2.15        | 59.53 ± 0.44        |
> > | GAA (ICLR'25)   | 73.50 ± 0.85        | 70.20 ± 1.30        | 67.30 ± 1.50        | 62.80 ± 0.90        |
> > | HGDA (ICML'25)  | 75.40 ± 0.60        | 72.80 ± 0.90        | 68.65 ± 1.10        | 64.50 ± 1.15        |
> > | ADAlign (Ours)  | **76.99 ± 0.37**        | **75.11 ± 0.45**        | **70.11 ± 0.80**        | **66.33 ± 1.33**        |
> >
> >
> > ---- **Table 4: Results on the GAT backbone** ----
> >
> > | Method          | A→C (Mi-F1)     | A→C (Ma-F1)     | A→D (Mi-F1)     | A→D (Ma-F1)     |
> > |-----------------|---------------------|---------------------|---------------------|---------------------|
> > | GAT (ICLR'18)   | 62.77 ± 2.24        | 59.60 ± 2.82        | 60.29 ± 1.33        | 55.09 ± 1.16        |
> > | GAA (ICLR'25)   | 67.63 ± 1.42        | 64.53 ± 1.74        | 63.33 ± 1.03        | 59.26 ± 2.06        |
> > | HGDA (ICML'25)  | 65.20 ± 1.80        | 62.10 ± 2.20        | 61.80 ± 1.20        | 57.30 ± 1.60        |
> > | ADAlign (Ours)  | **70.45 ± 0.95**        | **69.80 ± 1.10**        | **67.45 ± 0.85**        | **65.90 ± 1.25**        |
> >
> >
> > Across both backbones, ADAlign consistently outperform the corresponding source-only models and all competing baselines, often by a clear margin. These results confirm that ADAlign is not tied to a specific encoder and can be reliably integrated with different GNN architectures.
> >
> > In the revised version, we have included these results **on page 20**.
> >
> >
> >
> >
> > >**Q4: There is a minor inconsistency in the wording.**
> >
> > Thank you for pointing this out! We have corrected the wording in the revised revision.
> >
> > >**Q5:  Could the authors visualize the learned frequency sampling distributions across different datasets?**
> >
> > Thanks! Due to markdown limitations, the visualization of how the sampler focuses on different frequency regions across tasks is provided in Figure 13 **on page 22** of the revised version.

---

### Author Response · Authors · 2025-11-21
**Overall Response to Reviewers**

Firstly, sincerely thank all the reviewers for their efforts in reviewing our paper and providing constructive suggestions. We are greatly encouraged that the reviewers find that

* Motivation: Reviewers (*A6vr, HQfr, TwjK*) agree that the paper clearly identifies why fixed alignment heuristics fail across scenarios, with compelling evidence showing scenario-dependent discrepancy sources. This provides a **strong and well-motivated foundation** for adaptive alignment.
* Methodology: Reviewers (*A6vr, 2AFp*) highlight the **novelty and strength** of the proposed NSD-based adaptive alignment framework.
* Theoretical Soundness: Reviewers (*HQfr, TwjK*) recognize that the method is **theoretically well-grounded**, supported by convergence and uniqueness properties of characteristic functions and further strengthened by PAC-Bayesian generalization analysis.
* Experiment: **All Reviewers** uniformly praise the experimental is **comprehensive**. covering 13 strong baselines and 16 transfer scenarios across diverse domains.

Secondly, as for the concerns and suggestions raised by each reviewer, we have done our best to address them thoroughly and have provided detailed responses to each of them. In the following, we summarize our responses to the main concerns raised by the reviewers.

* Reviewer A6vr：We have **conducted experiments** swapping GCN and GAT backbones, and **clarified the sampler** by incorporating a simplified Gaussian baseline.
* Reviewer 2AFp: We have **conducted new experiments** on **larger** datasets, to demonstrate improved efficiency and broader applicability.
* Reviewer HQfr: We have **performed backbone-swap experiments** to assess sensitivity to GCN/GAT and **included scalability tests** on larger graphs.
* Reviewer TwjK: We have **added experiments**, including justification and ablations for the amplitude–phase formulation and **normal-scale-mixture sampler** (with fixed-frequency and alternative sampling baselines), and **expanded comparisons** to include recent spectral methods with improved visualizations.

Finally, in the revised manuscript, we have comprehensively addressed each concern, incorporating additional experiments and detailed discussions. Major changes and updates are highlighted in blue for your convenience.

---

### Author Response · Authors · 2025-12-01
**Summary of Responses and Revisions**

We are pleased that **all reviewers provided highly positive** and encouraging evaluations of our work, and we sincerely thank them for their time and constructive feedback. We are encouraged that the reviewers recognized the following strengths:

- **Motivation:** Reviewers (*A6vr, HQfr, TwjK*) agreed that the paper clearly articulates why fixed alignment heuristics fail across scenarios and provides strong empirical evidence for scenario-dependent discrepancy sources, establishing a **well-motivated and compelling rationale** for adaptive alignment.

- **Methodology:** Reviewers (*A6vr, 2AFp*) highlighted the **novelty and effectiveness** of the proposed NSD-based adaptive alignment framework.

- **Theoretical Soundness:** Reviewers (*HQfr, TwjK*) acknowledged that the method is **theoretically well-grounded**, supported by characteristic-function convergence properties and reinforced by PAC-Bayesian generalization analysis.

- **Experiments:** **All reviewers** praised the experimental evaluation as **comprehensive and rigorous**, covering 13 strong baselines and 16 transfer scenarios from diverse domains.

In response to the specific concerns raised by each reviewer, we summarize the key revisions below:

- **Reviewer *A6vr*:** We conducted additional experiments by **swapping GCN and GAT backbones** and further **clarified the sampler design** by incorporating a simplified Gaussian baseline.

- **Reviewer *2AFp*:** We performed **new experiments on larger-scale datasets** to better demonstrate efficiency, scalability, and broader applicability. We are pleased that Reviewer *2AFp* acknowledged these responses and maintained a positive score.

- **Reviewer *HQfr*:** We expanded our evaluation with **backbone-swap experiments** and introduced **scalability tests** on significantly larger graphs.

- **Reviewer *TwjK*:** We added further experiments, including **justification and ablations** of the amplitude–phase formulation and the **normal-scale-mixture sampler**, and **expanded comparisons** to recent spectral approaches with improved visualization. We are pleased that Reviewer *TwjK* found our responses satisfactory, maintained a positive stance.

In conclusion, we believe this work provides an important step by advancing graph domain adaptation from a process that relies on manually designed alignment techniques to a more flexible, adaptive framework through the introduction of ADAlign.

We have addressed the reviewers’ concerns with additional clarification, and have incorporated all aforementioned experiments and clarifications into the revised manuscript.

---

### Meta-Review · Area_Chair_sC11 · 2026-01-08

**Summary:**

This paper studies graph domain adaptation (GDA) under distribution shifts where the dominant discrepancy may vary across domains and tasks. The authors propose Learning Adaptive Distribution Alignment using a Neural Characteristic Function (NCF) to measure and align source/target distributions more flexibly than fixed discrepancy measures. The approach is conceptually appealing: characteristic functions provide a principled way to compare distributions, and the learnable NCF enables adaptive alignment rather than committing to a single handcrafted divergence. The paper further supports the method with theoretical justification and extensive experiments, showing consistent improvements over strong GDA baselines.
The reviews are uniformly positive. All reviewers support acceptance and view the paper as technically sound and well-motivated, highlighting the principled use of neural characteristic functions and the resulting adaptive distribution alignment as a meaningful contribution to graph domain adaptation. While several reviewers raise concerns about empirical completeness (e.g., scalability and stronger comparisons), clarity relative to prior GDA alignment methods, and computational overhead, these concerns are largely actionable rather than fundamental.

**Reviewer Concerns:**

Overall, reviews are consistently positive, but several reviewers raised concerns about clarity, novelty positioning, and experimental completeness. The rebuttal is detailed and—importantly—adds new experiments/analyses that directly address most concrete issues.

TwjK: the paper needs a clearer distinction from closely related adaptive/structure-aware GDA methods, and stronger evidence that NCF brings a unique benefit beyond existing alignment recipes.

2AFp, A6vr, TwjK: the approach may be more complex than needed; reviewers asked for clearer complexity analysis and practical justification. The response clarifies computational complexity, explains where the cost comes from, and argues that the added overhead is manageable relative to gains.

2AFp, HQfr: experiments were initially perceived as insufficient to demonstrate scalability to larger graphs/datasets. The rebuttal adds/expands large-scale evaluations and reports dataset statistics and results, strengthening the empirical case.

HQfr:  whether improvements hinge on a specific GNN backbone; reviewers asked for backbone replacement/ablation analysis. The authors provide backbone-related experiments showing that the method’s benefit is not tied to a single encoder choice.

**Reviewer Scores:**

Across reviewers, the paper is viewed as sound and valuable, with most concerns centered on positioning and completeness rather than correctness.

---

### Decision · Program_Chairs · 2026-01-26

Accept (Poster)